# GERALDINE (Google earth Engine supRaglAciaL Debris INput dEtector) - A new Tool for Identifying and Monitoring Supraglacial Landslide Inputs

William D. Smith[1], Stuart A. Dunning[1], Stephen Brough[1,2], Neil Ross[1], Jon Telling[3]

[1] School of Geography, Politics and Sociology, Newcastle University, Newcastle upon Tyne, UK.
[2] Department of Geography and Planning, School of Environmental Sciences, University of Liverpool, UK.
[3] School of Natural and Environmental Sciences, Newcastle University, Newcastle upon Tyne, UK.

*Correspondence to*: William D. Smith (w.d.smith2@newcastle.ac.uk)

**Abstract.** Landslides in glacial environments are high-magnitude, long runout events, believed to be increasing in frequency as a paraglacial response to ice-retreat/thinning, and arguably, due to warming temperatures/degrading permafrost above current glaciers. However, our ability to test these assumptions by quantifying the temporal sequencing of debris inputs over large spatial and temporal extents is limited in areas with glacier ice. Discrete landslide debris inputs, particularly in accumulation areas are rapidly 'lost', being reworked by motion and icefalls, and/or covered by snowfall. Although large landslides can be detected and located using their seismic signature, smaller ($M \leq 5.0$) landslides frequently go undetected because their seismic signature is less than the noise floor, particularly supraglacially deposited landslides which feature a "quiet" runout over snow. Here, we present GERALDINE (Google earth Engine supRaglAciaL Debris INput dEtector): a new free-to-use tool leveraging Landsat 4-8 satellite imagery and Google Earth Engine. GERALDINE outputs maps of new supraglacial debris additions within user-defined areas and time ranges, providing a user with a reference map, from which large debris inputs such as supraglacial landslides ($> 0.05$ km$^2$) can be rapidly identified. We validate the effectiveness of GERALDINE outputs using published supraglacial rock avalanche inventories, then demonstrate its potential by identifying two previously unknown, large ($>2$ km$^2$), landslide-derived supraglacial debris inputs onto glaciers in the Hayes Range, Alaska, one of which was not detected seismically. GERALDINE is a first step towards a complete global magnitude-frequency of landslide inputs onto glaciers over the 37 years of Landsat Thematic Mapper imagery.

## 1.0 Introduction

There are currently >200,000 glaciers worldwide covering >700,000 km$^2$, of which 8.2% are less than 1 km$^2$ (Herreid and Pellicciotti, 2020), excluding the Greenland and Antarctic ice sheets (RGI Consortium, 2017). Recent estimates suggest supraglacial debris only covers 7.3% of the area of this glacier (Herreid and Pellicciotti, 2020), up from 4.4% estimated by Scherler et al. (2018). However, for many glaciers it plays a critical role in controlling a glaciers response to climate change, due to its influence on surface ablation and mass loss (Benn et al., 2012; Mihalcea et al., 2008a, 2008b; Nicholson and Benn, 2006; Østrem, 1959; Reznichenko et al., 2010). Extensive debris coverage can alter the hydrological regime of a glacier (Fyffe et al., 2019), with the potential to increase/decrease downstream freshwater availability (Akhtar et al., 2008), and can play a key role in controlling rates of glacier thinning and/or recession, subsequently contributing to sea level rise (Berthier et al., 2010). This supraglacial debris control is thought to be increasingly important with more negative glacier mass balances, with retreating glaciers being increasingly characterised by expanding debris cover extents (Kirkbride and Deline, 2013; Scherler et al., 2011b; Tielidze et al., 2020). The expansion of supraglacial debris cover is due to: (i) glaciological and climatological controls such as thrusting and meltout of sub- and en-glacial sediment onto the surface (e.g. Kirkbride & Deline, 2013; Mackay et al., 2014; Wirbel et al., 2018); (ii) debris input from surrounding valley walls through bedrock mass movements (Deline et al., 2014; Porter et al., 2010); (iii) dispersion of medial moraines (Anderson, 2000); and, (iv) remobilisation of debris stores, particularly lateral moraines (Van Woerkom et al., 2019). The relative contributions of 'glacially' derived sediment, which

may in fact be the re-emergence of glacially modified mass movements (Mackay et al., 2014), as compared to direct subaerial inputs, is highly variable and there is complex coupling between hillslopes and glaciers that varies with relief (Scherler et al., 2011a). However, recent evidence from the Greater Caucasus region (Eurasia) suggests that supraglacially deposited rock avalanches (RAs), attributed to processes associated with climate change, are a key factor in increasing supraglacial debris coverage (Tielidze et al. 2020). Magnitude-frequency relationships suggest these low frequency, high magnitude events have a disproportionate effect on sediment delivery (Korup and Clague, 2009; Malamud et al., 2004). One of these large events mobilises enough debris to dominate overall volumetric production and delivery rates, exceeding that of the much higher frequency but lower magnitude events. Here we focus on supraglacial landslide deposits ($>0.05$ km$^2$), commonly associated with RAs, defined as landslides: (a) of high magnitude ($> 10^6$ m$^3$); (b) perceived low frequency; (c) long runout; and (d) where there is disparity between high present-day rates of slope processes above ice (Allen et al., 2011; Coe et al., 2018) and expected rates based on theories of lagged paraglacial slope responses (Ballantyne, 2002; Ballantyne et al., 2014a).

In formerly-glaciated landscapes, dating of RA deposits has shown a lagged response of paraglacial slope activity since deglaciation (Ballantyne et al., 2014b; Pánek et al., 2017). Events cluster in deep glacially eroded troughs and inner gorges at relatively low elevations in the landscape (Blöthe et al., 2015). Numerical modelling has shown how considerable rock-mass damage is possible during the first deglaciation cycle (Grämiger et al., 2017); some of the largest inventories highlight a close association with former glacier limits and the source zones of RAs, particularly in the vicinity of glacial breaches (Jarman and Harrison, 2019). However, almost all of our knowledge of past events relies on the presence of in-situ RA deposits. Due to erosional and depositional censuring such deposits are heavily biased to ice-free landscapes where rates of unmodified preservation are higher, although these are still unlikely to constrain true magnitude-frequencies unless rates of geomorphic turn-over are low (Sanhueza-Pino et al., 2011). In supraglacial settings, landslides, where topography allows, travel much further than their non-glacial counterparts due to the reduced friction of the ice surface (Sosio et al., 2012). Rapid transportation away from source areas also occurs because of glacier flow. This removes the simplest diagnostic evidence of a subaerial mass movement process – a linked bedrock source area and debris deposit. Without the associated deposit, bedrock source areas are easily mistaken as glacial cirques (Turnbull and Davies, 2006). Fresh snowfall or wind redistribution can rapidly cover a RA deposit that is many kilometres square in area (Dunning et al., 2015). If this occurs within the accumulation zone the deposit is essentially lost to all surface investigation and non-ice-penetrating remote sensing and ground-based techniques until eventual re-emergence in the ablation zone, after potentially considerable modification by transport processes. If a RA is deposited into the ablation zone, surficial visibility may be seasonal, but through time surface transport disrupts initially distinctive emplacement forms (Uhlmann et al., 2013). This supraglacial debris loading represents a glacier input (Jamieson et al., 2015) and can alter glacier mass balance, influence localised melt regimes (Hewitt, 2009; Reznichenko et al., 2011), and glacier velocity (Bhutiyani and Mahto, 2018; Shugar et al., 2012), leading to speed-ups and terminus positions asynchronous with current climatic conditions. Sometimes this leads to moraines that are out of phase with climate, due to the reduction in surface ablation and surging (or the slowing of a retreat) caused by large landslide inputs (Hewitt, 1999; Reznichenko et al., 2011; Shulmeister et al., 2009; Tovar et al., 2008; Vacco et al., 2010).

Currently, the detection of large supraglacially deposited landslides – other than through the most common form of ground-based detection, eye-witness reporting – is through the application of optical satellite imagery. This is a labour and previously computationally intensive process, often involving the downloading, pre-processing and manual analysis of large volumes (gigabytes) of satellite imagery. Manual imagery analysis to identify supraglacial landslide deposits and RAs has principally been applied in Alaska. This technique enabled detection of 123 supraglacial landslide deposits in the Chugach Mountains (Uhlmann et al., 2013), 24 RAs in Glacier Bay National Park (Coe et al., 2018), and more recently, 220 RAs in the St Elias Mountains (Bessette-Kirton and Coe, 2020). These studies acknowledge that their inventories are incomplete/underestimates

due to analysis of summer only imagery and an inability to detect events that are rapidly advected into the ice. These are critical drawbacks preventing accurate magnitude-frequency relationships from being derived, but analysis of more imagery over larger areas is unfeasible due to time and computational requirements. Studies of this kind are also typically in response to a trigger event e.g. earthquake or a cluster of large RA events (e.g. Coe et al. (2018) in Glacier Bay National Park), spatially biasing inventories into areas with known activity. They therefore provide a snapshot in time, with no continuous record. Methods are needed which are accessible, quick and easy to apply and require no specialist knowledge, to re-evaluate magnitude-frequencies in glacial environments. Currently, the only method capable of identifying a continuous record of such events, is seismic monitoring (Ekström and Stark, 2013). Seismic detection utilises the global seismic network to detect long-period surface waves, characteristic of seismogenic landslides. Seismic methods have identified some of the largest supraglacially deposited RAs in recent times (e.g. Lamplugh glacier RA (Dufresne et al., 2019)) which are compiled in a database (IRIS DMC, 2017), and, when combined with manual analysis of satellite imagery, gives information on duration, momenta, potential energy loss, mass and runout trajectory. However, landslides are challenging to detect using seismic methods and event positional accuracy is limited to a 20 – 100 km radius, due to the lack of high frequency waves when compared to earthquakes, further inhibited by the low frequencies and long wavelengths of dominant seismic waves worldwide (Ekström and Stark, 2013). This also results in an inability to detect landslides that are relatively low in volume, due to their weak seismic fingerprint ($M < 5.0$) and causes underestimation of landslide properties (e.g. event size and duration) because their runouts are seismically "quiet", likely due to frictional melting of glacier ice (Ekström and Stark, 2013). Despite these difficulties, current studies seem to indicate an increase in the rates of rock avalanching onto ice in rapidly deglaciating regions such as Alaska and the Southern Alps of New Zealand, where the majority of recent (aseismic) RAs are associated with glaciers. This increase has been linked to climate warming (Huggel et al., 2012) and potential feedbacks with permafrost degradation (Allen et al., 2009; Coe et al., 2018; Krautblatter et al., 2013). These links, coupled with the availability of high spatial and temporal resolution optical satellite imagery, have demonstrated the need for systematic observations of landslides in mountainous cryospheric environments (Coe, 2020). Five 'bellwether' sites have been suggested for these purposes: the Northern Patagonia Ice Field, Western European Alps, Eastern Karakorum in the Himalayas, Southern Alps of New Zealand and the Fairweather Range in Alaska (Coe, 2020).

The large archives of optical imagery, coupled with the recent boom in cloud-computing platforms, now provides the perfect combination of resources, which can be exploited to identify supraglacially deposited landslides on a large scale. Since the launch of Landsat 1 in July 1972, optical satellites have imaged the earth surface at increasing temporal and spatial frequency. Six successful Landsat missions have followed Landsat 1, making it the longest continuous optical imagery data series, revolutionising global land monitoring (Wulder et al., 2019). Analysis ready Landsat data is available for Landsat 4 (1982-1993), Landsat 5 (1984-2012), Landsat 7 (1999-present) and Landsat 8 (2013-present), providing 38 years of data at a 30 m spatial resolution and a 16-day temporal resolution. These data are categorised into three tiers: (1) Tier 1 data that is radiometrically and geometrically corrected (< 12 m root mean square error); (2) Tier 2 data which is of lower geodetic accuracy (> 12 m root mean square error); and (3) Real Time imagery, which is available immediately after capture but uses preliminary geolocation data and thermal bands require additional processing, before being moved to its final imagery tier (1 or 2) within 26 days for Landsat 7, and 16 days for Landsat 8. Traditionally, it has been difficult to exploit these extensive optical imagery collections such as Landsat, without vast amounts of computing resources. However, in the last decade, cloud computing has become increasingly accessible. This allows a user to manipulate and process data on remote servers, removing the need for a high-performance personal computer. Google Earth Engine (GEE) is a cloud platform created specifically to aid the analysis of planetary-scale geospatial datasets such as Landsat and is freely available for research and education purposes (Gorelick et al., 2017).

Here, we utilise Google Earth Engine (GEE), and the Landsat data archive of 38 years of optical imagery, to present the Google earth Engine supRaglAciaL Debris INput dEtector (GERALDINE). A free-to-use tool to automatically delimit new supraglacial debris inputs over large areas and timescales, which then allows for rapid user-backed verification of inputs from large landslides specifically. GERALDINE is designed to allow quantification of the spatial and temporal underreporting of supraglacial landslides. We describe the methods behind GERALDINE, verify tool outputs against known supraglacial rock avalanche inventories, and, finally demonstrate tool effectiveness by using it to find two new supraglacial landslides, one of which cannot be found in the seismic archives.

## 2.0    Method

GERALDINE exploits the capability and large data archive of GEE (Gorelick et al., 2017), with all processing and data held in the cloud, removing the need to download raw data. By default, it utilises Tier 1 Landsat imagery (30 m pixel resolution) that has been converted to top-of-atmosphere spectral reflectance (Chander et al., 2009), from 1984 – present, incorporating Landsat 4, 5, 7, and 8. GERALDINE also gives the user the following options: (i) to utilise Tier 2 Landsat imagery; and, (ii) to utilise Real Time Landsat imagery. Tier 2 imagery is valuable in regions where Tier 1 imagery is limited, e.g. Antarctica where there is a lack of ground control points for imagery geolocation. Real Time imagery is useful for rapid identification of landslide locations if a seismic signal has been detected but an exact location has not been identified. Landsat imagery is used in conjunction with the Randolph Glacier Inventory (RGI) version 6.0 (RGI Consortium, 2017). The RGI is a global dataset of glacier outlines excluding those of the Greenland and Antarctic ice sheets, digitised both automatically and manually based on satellite imagery and local topographic maps (Pfeffer et al., 2014). RGI glacier boundaries are delineated from images acquired between 1943 and 2014, potentially introducing errors into analysis due to outdated boundaries (Herreid and Pellicciotti, 2020; Scherler et al., 2018) (see Supplementary Information Section 1.0). However, this database represents the best worldwide glacier inventory available and shrinking ice as the dominant global pattern means the tool is occasionally running over ice-free terrain with null results rather than missing potential supraglacial debris inputs. Any updated version of the RGI will be incorporated when available. Additionally, the RGI can be replaced by the user with shapefiles of the Greenland and Antarctic ice sheets (v1.1 line 536 and 543), if analysis is required in these regions, or higher resolution (user defined) glacier outlines, if the RGI is deemed insufficient.

## 2.1 Overview of processing flow

GERALDINE gathers all Landsat images from the user-specified date range and all the images in the year preceding this user-specified date range, within the user-specified region of interest (ROI), creating two image collections within GEE. Users should note that smaller ROIs and annual/sub-annual date ranges increase processing speed, with processing slowing considerably with >800 Landsat images (~160-1500 GB of data). The software clips all images to the ROI, applies a cloud mask, and then delineates supraglacial debris cover from snow and ice. GERALDINE acquires the maximum debris extent from both image collections, creating two maximum debris mosaics, then subtracts these mosaics and clips them to the RGI v6.0 (or user defined area if not using RGI) to output a map. This map highlights debris within the user-specified time period that was not present in the preceding year, which we term 'new debris additions'. This map is viewable within a web browser as a layer in the map window. However, as it is calculated 'on-the-fly' (Gorelick et al., 2017), large areas can be slow to navigate. All files can be exported in GeoJSON (Georeferenced JavaScript Object Notation) format for further analysis, including to verify if detections are discrete landslide inputs. This is recommended for large ROIs. An overview of the workflow is presented in Fig. 1 and the detail for each step described in Sections 2.1.1–2.1.3.

### 2.1.1 Cloud masking

GERALDINE masks cloud cover using the GEE built-in 'simple cloud score' function (Housman et al. 2018). This pixel-wise cloud probability score allows fast and efficient identification of clouds, suitable for large-scale analysis (Housman et al., 2018) and has been previously applied and well-justified for use in glacial environments (Scherler et al., 2018). A 20% threshold is applied to every image, thereby excluding any pixel with a cloud score >20% from the image. We quantitatively evaluated this threshold to ensure optimum tool performance (see Supporting Information Section 2.0). Cloud shadow is not masked as it was found to have a minimal effect on the tool delineating debris from snow/ice whilst greatly increasing processing time.

### 2.1.2 NDSI

The Normalised Difference Snow Index (NDSI) is a ratio calculated using the green (0.52-0.6 $\lambda$) and SWIR (1.55-1.75 $\lambda$) bands. It helps distinguish snow/ice from other land cover (Hall et al., 1995) and excels at detecting ice where topographic shading is commonplace (Racoviteanu et al., 2008), due to high reflectance in the visible range and strong absorption in the SWIR range. GERALDINE applies the NDSI to all images and a threshold of 0.4 is used to create a binary image of supraglacial debris (<0.4) and snow/ice (≥0.4). This threshold has been utilised by studies in the Andes (e.g. Burns and Nolin, 2014) and Himalaya (e.g. Zhang et al., 2019), but optimum thresholds often vary between 0.5 (Gjermundsen et al., 2011) and 0.2 (Keshri et al., 2009; Kraaijenbrink et al., 2017). We justify our 0.4 threshold based on Scherler et al. (2018) who deemed it optimum for the creation of a global supraglacial debris cover map using Landsat images. We advise users to use this default threshold but if this appears sub-optimum in a user defined region of interest (ROI), the threshold can be fine-tuned in the code (v1.1 line 244 and 254). We utilise NDSI instead of newer band ratio techniques (e.g. Keshri et al., 2009) and more complex algorithms (e.g. Bhardwaj et al., 2015) to ensure transferability between Landsat TM, ETM+ and OLI TIRS sensors as we wish to harness the full temporal archive.

### 2.1.3 Retrieving maximum debris extent

To attain a maximum debris extent, GERALDINE reduces each image collection to an individual image using a pixel-based approach (Fig. 2). Every binary image (supraglacial debris: 0, snow/ice: 1) in each image collection is stacked, with pixels in the same geographic location stacked sequentially. If any pixel in the temporal image stack is debris, the corresponding pixel in the final mosaic will be a debris pixel, creating a maximum debris extent mosaic. GERALDINE is therefore debris biased due to this processing step (Fig. 2). Calculated maximum debris extent mosaics for both the user-defined time period and previous year are differenced, the output being new debris additions. Both the previous year maximum debris extent, and new debris addition mosaics, are displayed for user analysis within the GEE interactive development environment, and easily exportable to Google Drive (included as part of sign-up to Google Earth Engine).

### 2.2 Validation

A two-part validation was undertaken to assess the effectiveness of GERALDINE outputs for allowing a user to rapidly identify supraglacially deposited landslides: a detection validation (i.e. can the user confirm a supraglacially deposited landslide has occurred from a GERALDINE output?), and an area validation (i.e. how much of the area of the supraglacial landslide deposit has GERALDINE detected?). Although areal detection is not the main purpose of the tool, greater area detection would ultimately help the user with identification of supraglacially deposited landslides. Validation was performed against the already-defined RA databases of Bessette-Kirton and Coe (2016), Deline et al. (2014), Uhlmann et al. (2013) and the Exotic Seismic Events Catalog (IRIS DMC, 2017). To provide validation, RAs had to occur after 1984 (onset of Landsat TM era) and had to deposit debris predominantly onto clean-ice areas of glaciers in the RGI. Forty-eight events out of a total of 325 met

these criteria, their locations distributed across the European Alps, Alaska, New Zealand, Canada, Russia and Iceland (Fig. S5).

GERALDINE was run for the year of the event using Landsat Tier 1 imagery; the new debris vector output file was exported into a GIS and after an initial qualitative step to see if the user would flag the RA from the GERALDINE output, the area of the deposit it detected was calculated within the GIS. We utilised the select by location tool in QGIS, to select any pixels/pixel clusters within/intersecting an outline of the RA manually-digitised from a Landsat image using the Google Earth Engine Digitisation Tool (GEEDiT) (Lea, 2018). We clipped selected pixels to the manually digitised RA outline and calculated the area of these selected pixels. The tool-detected area was then compared against the area of the manually digitised RA outline. These two steps allow for an assessment of GERALDINEs ability to highlight new debris inputs, and if this changes over the Landsat era.

### 3.0    Results and Discussion

### 3.1 Validation

Of the 48 validation RAs, the user was able to correctly identify 44 of these events from GERALDINE output maps, a true positive detection accuracy of 92 %. False negatives all pre-date 1991 (Fig. 3), giving 100% successful user identification post-1991. These false negatives can be explained by a failure of Landsat satellites from imaging the RA deposit. This was due to reduced (and insufficient in this case) Tier 1 Landsat image availability pre-Landsat 7 within the GEE data catalogue, inhibiting GERALDINE from highlighting the RA as new debris. We note that if just one image featured the RA, GERALDINE would highlight the deposit as new debris due to its bias towards debris detection (see section 2.1.3). However, a true 100 % detection rate for supraglacial landslide deposits on glaciers is unlikely, due to some deposits running out over existing debris cover, and some having high snow/ice content or entraining large amounts of snow/ice during events, which can be common for landslides deposited supraglacially. This high snow/ice content can mask them as snow/ice during NDSI delineation from debris, inhibiting detection. However, events of this kind also pose significant difficulty for user delineation with original optical imagery. GERALDINE works best when a number of images in the image stack represent maximal debris cover in the preceding year, reducing false positives for the timespan of interest i.e. flagging old debris as new debris, due to a lack of old debris exposure in the previous year. This is particularly applicable to small (<0.5 km$^2$) glaciers, where the overall significance of a single pixel increases. The debris bias of GERALDINE ensures true negative detection is also extremely high, but this high true negative detection is why user verification of new debris outputs is needed, because they are flagged as new debris but display no supraglacial RA characteristics i.e. lobate and elongated (Deline et al., 2014). To a user familiar with glacial and landslide processes, the differences in GERALDINE outputs between true positives/negatives and false positives/negatives are clear when running the tool to find RA inputs.

GERALDINE RA areal accuracy increases over time from 19 % in the Landsat 4/5 era, to 71 % with the current Landsat 7/8 constellation (Fig. 3), with the latter period characterised by increasingly modern sensors with greater spectral and temporal resolution. Low areal accuracy in the Landsat 4/5 era is once again a product of the GEE data catalogue having limited imagery for certain years in glaciated areas, reducing the ability of GERALDINE to detect the entire area of new debris additions. Areal accuracy increases after the failure of Landsat 4 in December 1993, at which point Landsat 5 is the sole data collector of imagery at a frequency of every 16 days. Despite this single functioning satellite, the tool detects all eight validation events and on average 59 % of the deposit areas between 1993 and the activation of Landsat 7 in 1999. The dual Landsat 5/7 constellation increases tool area accuracy further to 69 %. However, a decrease in mean area accuracy is evident after the failure of the Landsat 7 Scan Line Corrector in May 2003 (Markham et al., 2004), decreasing tool areal accuracy by 4 %, due to images missing up to 20-25 % of data per image in the stack (Hossain et al., 2015). We find that a number of Landsat 7

scenes also feature stripes of no data, pre-dating the scan line corrector failure, and can inaccurately cause 'stripes' of new debris in tool outputs. The current Landsat 7/8 constellation has the highest accuracy for detecting the area of RAs at 71 %. The smallest new debris addition we used for validation was 0.062 km$^2$, of which GERALDINE detected 71 % of the area, so we have confidence in detection greater than 0.05 km$^2$, equating to ~56 Landsat pixels. Even with GERALDINE performing well, additional refinement and/or full automation of landslide deposit identification would be an interesting, and priority, area for further investigation. We also envisage development with other higher resolution and higher repeat satellites e.g. the Sentinel 2 and Planet Lab constellations. However, we found that current cloud mask algorithms for these data are not sufficient for accurate global glacial debris delineation.

GERALDINE is frequently affected by the RGI dataset causing over/under-estimation of previous year debris extents and new debris additions. For example, at tidewater glaciers that have undergone retreat since their margins were digitised, the tool often detects clean ice and debris at the tongue. This is dependent on the presence of ice mélange (NDSI classification as ice/snow) and dark fjord water (NDSI misclassification as debris) in imagery (see Supplementary Information Section 1.0). In addition, we found an instance where a supraglacial landslide deposit had been misclassified as a nunatak (60°27'23.7"N, 142°33'35.7"W) and therefore this section of the glacier is erroneously missing from the RGI dataset altogether, preventing tool detection, but this is likely a single case. Topographic shading and/or bright illumination of debris cover can at times cause pixels to be masked from Landsat scenes due to misclassification as cloud (see Supplementary Information Section 2.0); however, if the tool is run over a sufficiently long period, this will not influence new debris detection. GERALDINE can also not detect landslide debris deposition onto an existing debris cover. Therefore, if a landslide consists of multiple failures, a GERALDINE output map would only detect one event, with the deposit extent being the combined total of all failures. It would be highly beneficial to combine GERALDINE with seismic detection to help delineate the amount of failures that occur.

### 3.2 New Supraglacial Landslide Input Detection Example

The Hayes Range, Alaska has a history of large supraglacial debris additions (e.g. Jibson et al., 2006), but no events have been documented in the last decade, in contrast to a recent dense cluster in the Glacier Bay area of Alaska (Coe et al., 2018), which formed part of the validation dataset. To test this, we ran GERALDINE for 2018 to highlight new debris additions on glaciers in the Hayes Range (Fig. 4a). GERALDINE used a total of 228 Landsat images for analysis; 107 to determine the 2017 debris extent and 121 to determine the 2018 debris extent. Landsat tiles vary from 200 MB to 1000 MB when compressed, so, if we assume an average tile is 500 MB, a user would require 114 GB of local storage, a large bandwidth internet connection to download (which comes with an associated carbon cost), and, a PC capable of processing these data. GEE required none of these requirements and completed analysis in under two minutes, extracting information from every available cloud-free pixel, to maximise use of the imagery. The new debris output map produced was 6.5 MB, and contained all relevant 'new' debris information from 2018. The output map highlighted two large supraglacial landslide deposits, which occurred between 1 January 2018 and 31 December 2018. These were manually verified and the potential window of event occurrence identified using satellite imagery within GeeDiT (Lea, 2018). The larger of the two deposits is from a slope collapse on the southern flank of Mt Hayes (4216 m) (63°35'11.7"N, 146°42'50.0"W), with emplacement determined between 10 and 25 February 2018 (Fig. 4b). This supraglacial landslide was also detected using the seismic method (Ekström and Stark, 2013 see Section 1.0), and confirmed as occurring on 12 February 2018 (Goran Ekström, personal communication, 2019). The resulting debris deposit covered 9.4 km$^2$ of the surface of the Susitna Glacier (digitised from Planet Labs Inc. imagery from 31/07/2018). The tool detected 27.5 % of the area of this deposit, due to emplacement predominantly in the accumulation area, with the upper half of the deposit rapidly covered by snow after the event. The second, smaller supraglacial landslide deposit occurred between 4 and 7 July 2018, on an unnamed glacier to the east of Maclaren Glacier (63°20'21.9"N, 146°26'36.1"W) (Fig. 4c). GERALDINE detected 78 % of this 1.9 km$^2$ supraglacial debris input, which transformed the glacier from 16 % debris covered

to 51 % debris covered, and will have important implications for glacier melt regime, velocity and response to atmospheric drivers. Unlike the larger supraglacially deposited landslide from Mt Hayes, this event was not automatically detected using seismic methods (Goran Ekström, personal communication, 2019), suggesting that its seismic signature was lower than the seismic detection limit (M < 5.0) (Ekström and Stark, 2013). Therefore, there is a high potential to detect all events using GERALDINE, and then provide time-location filters to seismic records to retrospectively quantify force histories and precise timings of events not flagged automatically as a landslide.

We note that new large debris inputs are partially highlighted on the Black Rapids Glacier for 2018 (Fig. 4d), but these 'new' additions were actually deposited in 2002 during the Denali earthquake (Jibson et al., 2006; Shugar et al., 2012; Shugar and Clague, 2011). We assign this discrepancy to minimal cloud-free imagery during summer (a time when deposits are uncovered by snow melt), preventing the tool from highlighting their full summer extent, and causing underestimation of the 2017 debris cover. To a human operator, however, it is clear these debris additions are erroneous because 'new' debris is patchy, with 2017 debris extent and snow/ice preventing detection of a homogeneous deposit. If GERALDINE is run annually for multiple years, the user will be able to determine the emplacement date for these earlier supraglacial landslide deposits.

### 3.3 Tracking new debris transportation

A secondary use of GERALDINE is tracking existing supraglacial landslide deposits. These deposits are transported down-glacier by ice flow, although often the initial emplacement geometry is characteristically deformed and spread due to differential ablation and ice motion (Reznichenko et al., 2011; Uhlmann et al., 2013). GERALDINE can give an indication of deposit behaviour and movement by highlighting 'new' debris, at the lateral and down-glacier end of the deposit, as it moves between image captures (Fig. 5). Differencing the distance of this new debris from the previous year's deposit extent can give an approximation of lateral spreading and glacier velocity over the user-specified time period, the latter of which is often unknown at the temporal resolution of Landsat and complex to calculate in high mountain regions (Sam et al., 2015).

To demonstrate the evolution of a RA through time, we ran GERALDINE for 2012, 2013, and 2014 for the Lituya Mountain RA in Alaska. This RA occurred on 11 June 2012 and was deposited onto a tributary of the John Hopkins glacier (Geertsema, 2012). The upper portion of the deposit was sequestered into the ice after its deposition in 2012, as is common of debris inputs in glacier accumulation areas (Dunning et al., 2015). However, the deposit toe remained visible on the surface, likely because it was below the snow line. We estimate the down-glacier transport velocity of this RA by tracking and measuring the movement of the deposit toe, to measure the displacement of the deposit leading edge. Using this method, estimates of down-glacier transportation of the deposit leading edge between 2012 and 2013 are ~575 ± 30 m, and ~328 ± 30 m between 2013 and 2014 (Fig. 5), the latter in agreement with glacier velocity calculated by Burgess et al. (2013) between 2007 and 2010 (250 – 350 m a$^{-1}$), and ITS_LIVE velocity from 2013 (300-400 m a$^{-1}$) (Gardner et al., 2018; Gardner et al., 2019). We suggest that the higher RA deposit velocities between 2012 and 2013 are a result of the immediate response of the glacier to reduced ablation rates directly beneath the debris, causing an ice-pedestal to form, from which debris is redistributed through avalanching off the pedestal sides, expanding debris coverage (Reznichenko et al., 2011). We note other areas are flagged as 'new debris' in 2013 and 2014. These are typically where glacier downwasting has occurred exposing more of the valley walls, or where there has been temporal evolution of the debris cover e.g. glacier flowline instabilities. These flow instabilities can cause double-counting of debris when larger time windows are specified (see Herreid and Truffer, 2016). Both processes subsequently cause false classification as 'new debris'. However, neither glacier downwasting nor evolution of the debris cover display supraglacial landslide characteristics, so it is highly unlikely that a user would mistake them for one.

## 4.0 Conclusion

GERALDINE is the first free-to-use resource that can rapidly highlight new supraglacial landslide deposits onto clean ice for a user-specified time and location. It can aggregate hundreds of Landsat images, utilising every available cloud-free pixel, to create maps of new supraglacial debris additions. Using the output maps produced, GERALDINE gives an objective starting point from which a user can identify new debris inputs, eliminating the time-intensive process of manually downloading, processing and inspecting numerous satellite images. The method allows user identification of mass movements deposited in glacier accumulation zones, which have very short residence times due to rapid advection into the ice. This is a process that has not previously been quantified. We demonstrate its effectiveness by verifying it against 48 known, large, supraglacially deposited rock avalanches that occurred in North America, Europe, Asia, and New Zealand. GERALDINE outputs helped identify 92% of all 48 events, with 100% successful identification post-1991 when image quality and availability increases. We showcase how GERALDINE does not suffer from the traditional disadvantages of current manual and seismic detection methods that can cause supraglacial landslides to go undetected, by identifying two new supraglacial landslides in 2018, in the Hayes Range of Alaska. One of these events was not detected using existing methods, therefore, the frequency of large supraglacial debris inputs is likely historically underestimated. We suggest users should apply GERALDINE at standardised time intervals in recently identified 'bellwether sites' in glaciated high mountain areas undergoing rapid change i.e. Greenland, Alaska, Patagonia, the European Alps, New Zealand Alps and the Himalaya, to investigate annual rates of these large debris inputs. GERALDINE can become part of the repertoire of tools that enable glacial landslides/rock avalanches to be identified in the past, present, and future. It will improve remote detection and characterisation of these events, to help quantify and evaluate their frequency, spatial distribution and long-term behaviour in a changing climate.

**Code/data availability**

GERALDINE code and the validation dataset are available at https://doi.org/10.5281/zenodo.3524414. All other results can be recreated by running GERALDINE in the respective example areas. A guide on how to use GERALDINE is provided in Supplementary Information Section 4.0.

**Author responsibilities**

WS developed the tool and wrote the manuscript. SD made substantial contributions to the conception and functionality of the tool, as well as manuscript editing. NR, SB and JT provided useful guidance on tool functionality and contributed to the manuscript.

**Acknowledgments**

This work was funded by a Newcastle University Research Excellence Award PhD scholarship to W. Smith. We acknowledge the freely available Landsat datasets provided by the USGS and hosted in the Google Earth Engine data catalogue, and the Randolph Glacier Inventory v6.0 (RGI Consortium, 2017). Hillshade IFSAR DTMs used for figure production were collected as part of the Alaska Mapping Initiative (doi:10.5066/P9C064CO) and are available through the USGS EarthExplorer data portal. We would also like to thank Ryan Dick for his feedback on early versions of GERALDINE.

**Competing interests**

The authors declare that they have no conflict of interests.

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

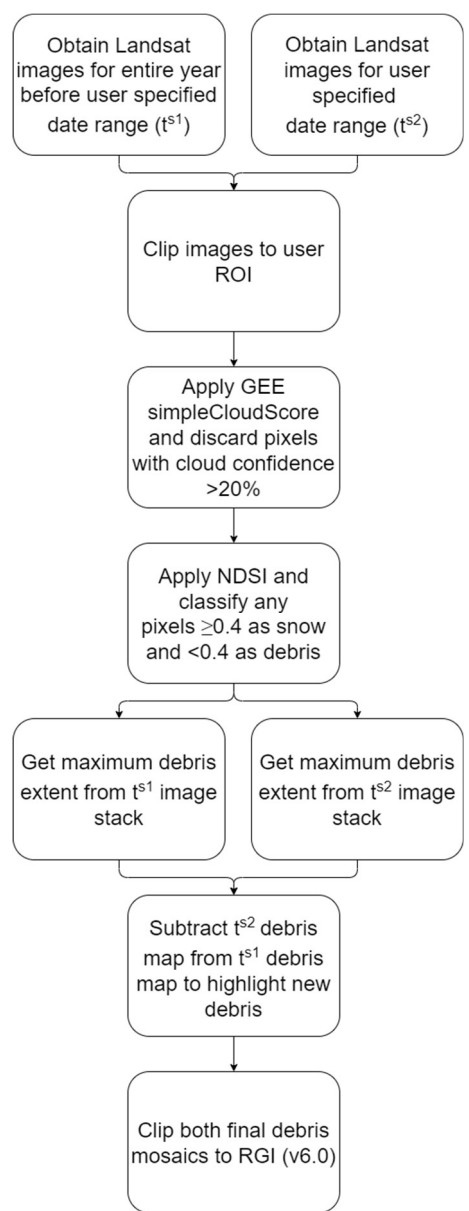

**Figure 1: Processing flow of GERALDINE.**

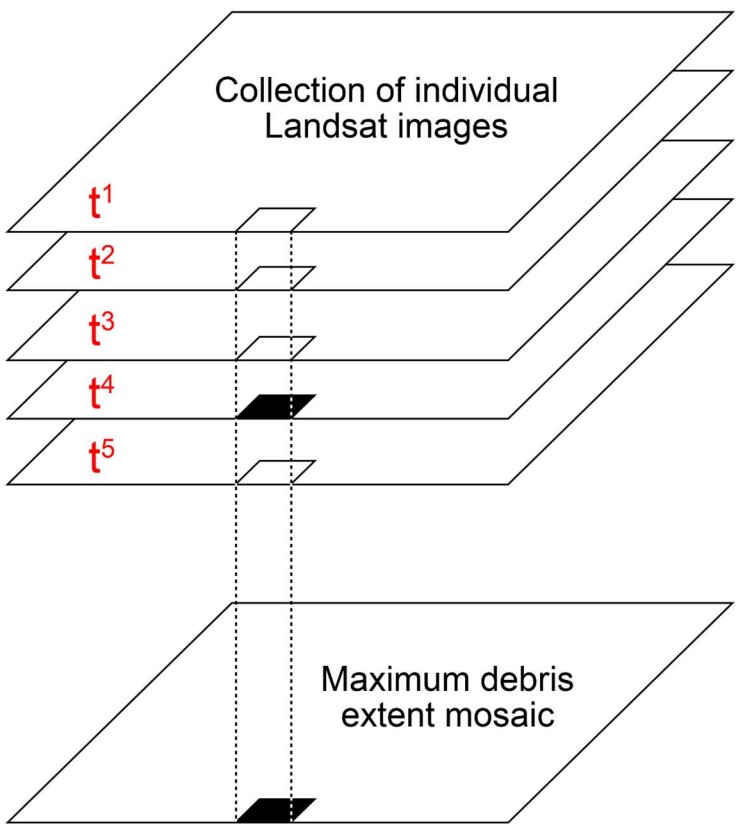

**Figure 2: Reducer diagram - GEE stacks all images in the collection and undertakes pixel-wise analysis of debris cover, to create a mosaic of maximum debris cover extent. If just one pixel in the image stack is debris, then the corresponding pixel in the maximum debris mosaic will be debris. White pixels represent snow/ice, black pixels represent debris.**

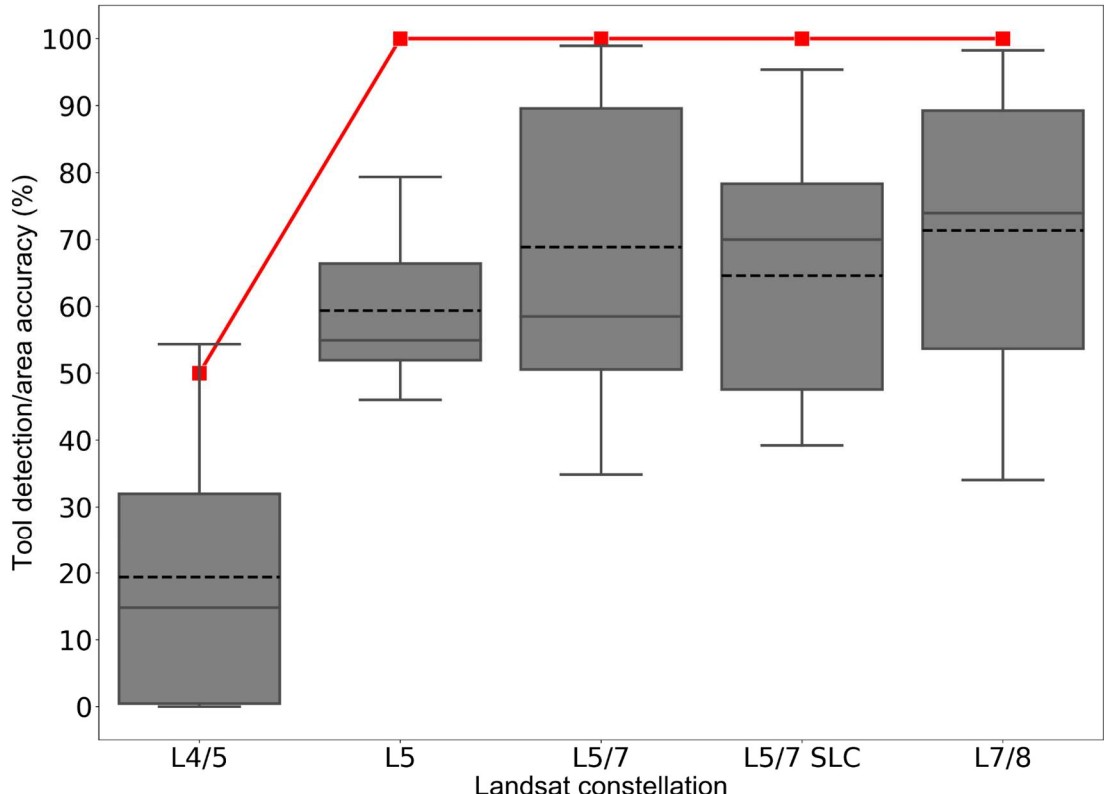

**Figure 3: GERALDINE rock avalanche (RA) detection accuracy (red line) and RA area accuracy (boxplots) with different Landsat constellations over time. L4/5 (1984-1993) – 8 validation RAs, L5 (1993-1999) – 8 validation RAs, L5/7 (1999-2003) – 9 validation RAs, L5/7 SLC (Scan Line Corrector failure) (2003-2013) – 11 validation RAs, and L7/8 (2013-present) – 12 validation RAs. Dashed line represents mean, solid line median, box represents upper and lower quartiles, whiskers represents min and max area accuracies.**

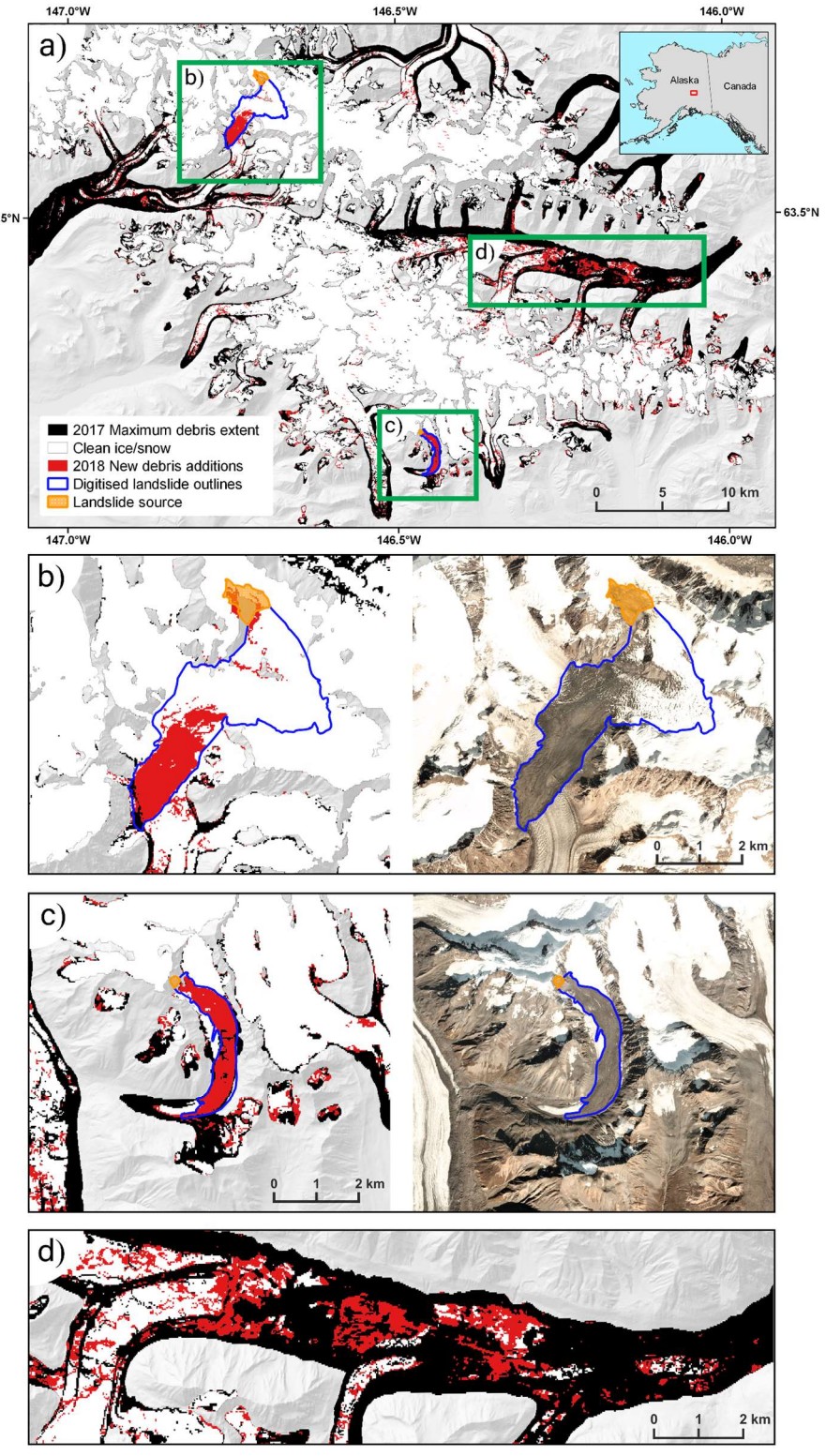

**Figure 4: a) 2018 new debris additions in the Hayes Range, Alaska. RA outlines digitised using Landsat imagery and the GEEDiT tool (Lea, 2018). Inset map denotes location of Hayes Range. b) GERALDINE output of Mt Hayes landslide extent and corresponding image courtesy of Planet Labs, Inc. (31/07/2018). c) GERALDINE output of landslide extent on a small valley glacier east of Maclaren glacier and corresponding image courtesy of Planet Labs, Inc. (13/09/2018). d) Erroneous 2018 tool detection of Black Rapids glacier RA deposits, which were deposited as a cause of the 2002 Denali earthquake (Jibson et al., 2006). Green boxes signify areas of interest and correspond to magnified areas of b), c) and d), respectively. IFSAR DTM background from the Alaska Mapping Initiative (doi: 10.5066/P9C064CO)**

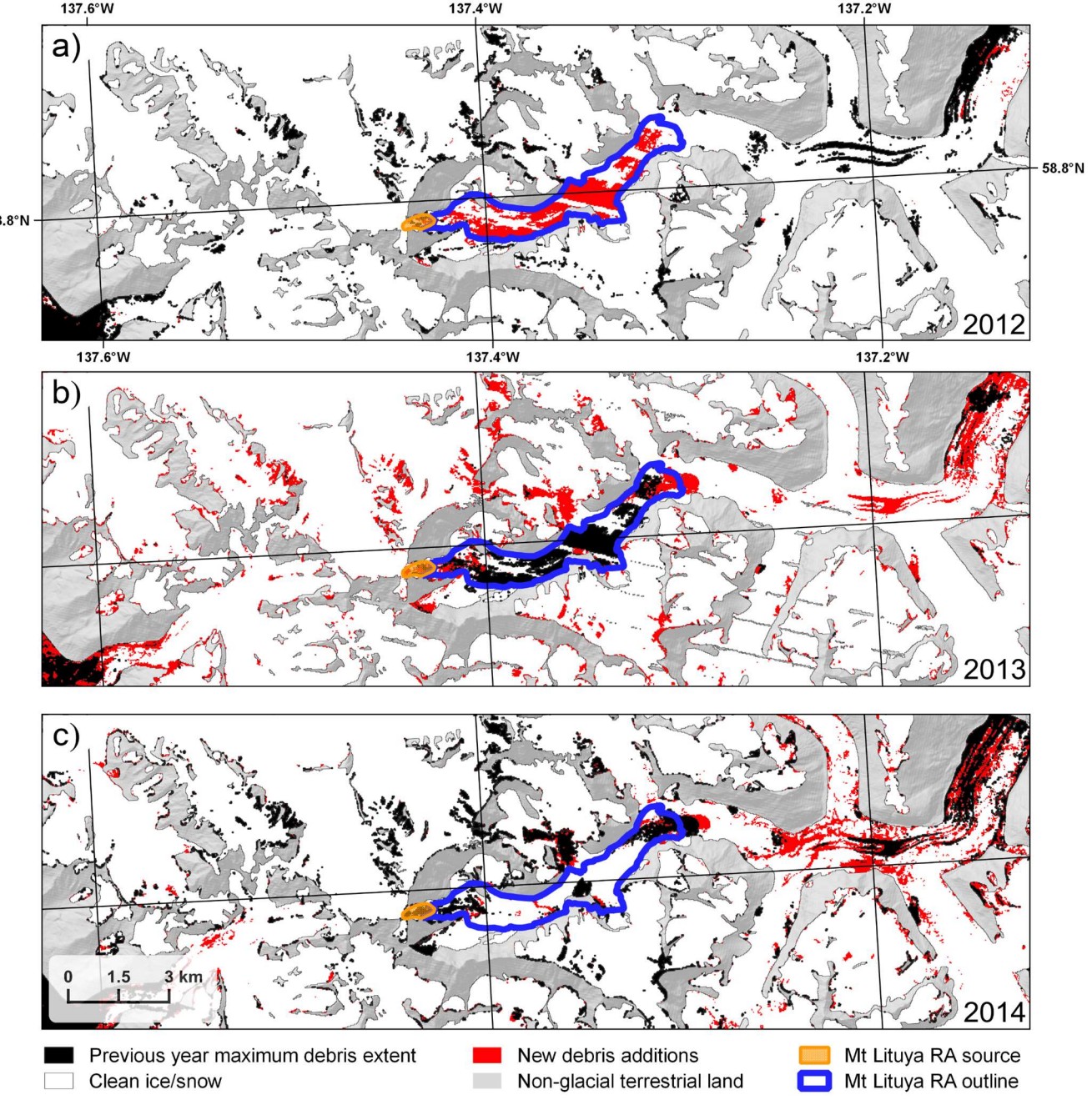

**Figure 5: Deposition and behaviour of Lituya RA, John Hopkins Glacier Alaska (58°48'54.3"N, 137°17'40.9"W) detected by GERALDINE when run for a) 2012, b) 2013, and c) 2014. Landsat 7 scan line corrector issue visible in lower right section of 2013 image (B). IFSAR DTM background from the Alaska Mapping Initiative (doi: 10.5066/P9C064CO).**