# Peer review of "GERALDINE (Google earth Engine supRaglAciaL Debris INput dEtector) - A new Tool for Identifying and Monitoring Supraglacial Landslide Inputs"

_Earth Surface Dynamics, 2020_

## Referee Comment (RC1) · Gioachino Roberti (Referee) · 4 Jul 2020

I enjoyed reading the paper "GERALDINE (Google earth Engine supRaglAciaL Debris INputdEtector) - A new Tool for Identifying and Monitoring Supraglacial Landslide Inputs" and I recommend it for publication. The paper presents a new tool to exploit Landsat images in Google Earth Engine to map debris onto glaciers, therefore providing a semi-automatic tool to identify rock avalanches emplaced on to glaciers, and to track supraglacial debris movement. This tool can complement seismic analysis, and, if extensively applied, help developing F-M curves of rock avalanches onto glaciers in

the past 37 years.

The following comments can help to further improve the paper.

I think a better overview of satellite spatial resolution and detectable landslide size is needed in introduction.

In the discussion section you could add some paragraphs:

1) A paragraph about development F-M curves, as it is a topic mentioned in introduction and conclusion but not directly addressed in the discussion

2) A paragraph about "eliminating the time-intensive process of manually downloading, processing and inspecting numerous satellite images" that is then mentioned in the conclusion. With considerations about transferability of the method to other satellites and data storage and processing platforms

3) and (eventually) how a similar approach may be used in other context (landslides in forested areas etc)

In line comments:

Line 14: Quantify? What's the size of the smallest detectable landslide?

Line 18: You can detect only the large landslides? This sounds in contrast with your earlier statement

Line 21: Ok cool. So large landslides that may not have been identified seismically. I don't think you need to try to "sell it" as alternative or better method than seismic identification, they can be used together. A tool like the one you have developed is cool by itself, seismic identification or not.

Line 24: Very cool! But you should expand the 37 year F-M topic in the discussions

Line 38: Can you method distinguish these 3 types of debris cover?

Line 40: How do you distinguish re-emerged debris vs supraglacially emplaced debris?

Line 43: Ok, you should be clearer about the landslide size in the abstract too

Line 80: How small is "smaller landslides"

Line 82: It sounds like you are detecting large landslides that have no seismic signature rather than "small landslides".. maybe you can reword a bit to put emphasis on the combination of size, frictional melting etc.

Line 89: It may be worth expanding this paragraph/add new paragraph and give an overview of GEE and Landsat satellites. I see you discuss landsat satellites in method and validation sections but an overview of the satellites (different tiers, spatial resolution, accuracy, years of operation and revisiting time etc..) in the intro will help the reader.

Line 97: I think the resolution should be mention earlier in the paper too

Line 197: Would this still be usefull to assess some of the other supraglacial debris types presented in the introduction? Expand on this (see my comment of figure 5).

Line 217: maybe you could also briefly discuss how this method could be applied (maybe not in GEE but in some other environment) to other satellites

Line 276: In the intro you mentioned the possibility of the development of frequency-magnitude curves for landslides onto glaciers, but there is not discussion of that point here. Maybe you can add a short paragraph (with example?) to explore that potential.

Line 277 Something else that feels like may be missing in the discussion is the overview of the value of "eliminating the time-intensive process of manually downloading, processing and inspecting numerous satellite images" that is then mentioned in the conclusion

Line 287: I agree, but you should discuss this in the discussion. See my comment about frequency-magnitude

Figure 4: Can you can mark the collapse scar of these landslides? will help the reader

Figure 5: Same, where is the landslide coming from? Can you discuss the origin of the other debris addition in the glacier on the top right of the picture? in relation to my comment at Line 197.

Very cool! Gio

---

## Referee Comment (RC2) · Michelle Koutnik (Referee) · 6 Jul 2020

In this study the authors develop a powerful new tool to identify supraglacial landslides. They present the tool, as well as demonstrate how it can work and the value of it by identifying two previously unknown landslide debris events. This is an exciting development, and valuable to capturing these events where evidence of them is often lost quickly on the landscape, and yet they are important debris sources. I enjoyed reading the paper and I really enjoyed thinking about what may be possible using the tool. I have some questions and suggestions for the authors, but all of these points are minor.

[Figure]

Overall, great work on this.

1) It could be worthwhile to put the size of landslide deposits that you can identify in more context with the size of glaciers that you can reliably search over and/or something about the size distribution of glaciers around the world. You don't have to answer this but it made me curious: over what proportion of the total number of glaciers would be possible to detect a rock avalanche of the size that you search, assuming that an event occurred? Is this the same as the base number of glaciers with debris mentioned, which was 4.4% of 215,547 glaciers worldwide?

And, the abstract mentions >2km2 area but around line 43 the mention is volume. It would be helpful to relate these together and also indicate how volumes are estimated. With respect to the events it may also help to explain why these are referred to as 'high magnitude' - is this your designation?

2) Another question is if this tool could detect smaller-scale events. Is it that any smaller events are not considered rock avalanches and/or that they cannot be detected? (I thought that "rock avalanches" were defined being >1Mm3, but I could be wrong about that)

What about rock avalanche events on already heavily debris-covered glacier surfaces - would those be detectable?

3) It could also be worthwhile putting the need / value of this tool in context with the total number of rock avalanches of this scale that have been found to date. Was the validation set of 48 known events chosen to span as many regions as possible, or are these all of the events that have been catalogued to date? More context on the likelihood to find additional, unidentified events would be helpful. Another way to expand on that could be to illustrate just how labor intensive it would be to search the Landsat archive manually. What is the range of repeat times of Landsat? This would also help put in context the two new events that you did identify.

Related to this point: I may have missed it, but how computationally and user-labor intensive is applying this tool. It sounds well beyond the scope of what a team like yours could do, but how far from possible would it be to search all glaciers where events may have occurred for the past 37 years? Is the challenge on the GEE computation side or on the validation side? When the latest RGI outlines come out is this something that could be done?

It may be your goal to let the curiosity of the users take over here, but are there some outcomes you think this makes possible in the short term and would advocate for (or may be doing yourselves?). Not having a sense of how intensive the process is, I was left to wonder the scope of study that may be reasonable to undertake - maybe in the conclusions you could indicate something about studies that seem worthwhile? For example, is this best applied to target regions of a certain size and/ or over target timeframes of a certain duration?

In the supplement it was mentioned that you compared to a Planet image. Is there anything that can be said about the future of applying this tool to other image sets? I understand that Planet images are not openly available, but is Landsat the only archive that makes sense to use? Is there anything to say about coordinating Landsat-based results with other image sets, or does that just need to be taken on a glacier-by-glacier basis? This would be important to at least mention, but doesn't take away from the achievement of getting this to work for Landsat data.

4) This is a subtle point, but it seemed like one that was done deliberately in the text so wanted to raise my reaction. The title (and acronym for the tool) uses "identifying" to describe what is done by the tool. And, the tool is referred to as a "detector". However, typically the text refers to what the tool provides as "highlighting" new events. It is only after user evaluation that they are "identified". If this was deliberate then I would check for complete consistency and maybe say that directly somewhere. I suggest that identify (or detect) is a reasonable term for what the tool does, and then the user confirms or validates that finding - but, any word choices you prefer will work as long as explained

clearly and used consistently. (Pay particular attention to this in the conclusion where the language seems to be mixed.)

Another subtle point on language is if all "supraglacial landslide inputs" are the same as "rock avalanches"? And, assuming that debris inputs are also the same thing? I would be check over to be clear and consistent.

Specific points:

It may be worth mentioning in the main text that updates to RGI can be readily accommodated. I have seen at least one announcement that RGI v7.0 has a release target by the end of 2020. This is indicated in the supplement but not stated directly (but maybe it is obvious).

Line 35: Consider referring to point (i) as glaciological and climatological controls?

Line 55: "rapidly transported away from source areas" - in addition to rapid sequestration, which is I think the point focused on in the sentence following the one where this is mentioned, is there a citation about how runout extent of the event is different when deposited primarily on ice?

Line 60: Why use the term "censoring" here?

Line 108: I would change this from "present day" since the RGI v6.0 was published in 2014 and likely stops with digitized outlines before then

Line 153: I'm not sure I understand the point that "GERALDINE is in effect standardised with this global supraglacial cover map" - it would be help to expand on this point

Line 185: I had to read this sentence a few times. Maybe stating this in terms of candidate events (instead of outputs) or being clear that identification step is the one that the user executes and that GERALDINE only presents candidates? (See point above, as I'm advocating for a particular language choice, just that it is a bit more clear and consistent)

Line 211: Is introducing the acronym SLC necessary? It is only used once (I think). In general there are a lot of acronyms (see comment below on Figure 1)

Line 255: Am I understanding this right that GERALDINE could not detect multiple landslide deposits in about the same spot but at different times? This may never (or only rarely) occur, but I wasn't sure if that was the point this sentence was trying to make. Or, something else about how the "user will have already determined the date of these earlier supraglacial landslides"

Section 3.3 - it seems like it would be worth mentioning that you can do this in the abstract. That would also help expand on the "monitoring" side of the tool's name up front

Line 285: What are the current methods that GERALDINE outperforms? Manual inspection of individual images?

- Very very minor, but I also found that the original (and widely cited) paper by Ostrem has his last name typically spelled with a slashed O but in the original paper it is given with an umlaut. From my reading of this it may have been an older alphabet choice and that these are the same (https://en.wikipedia.org/wiki/%C3%96; https://en.wikipedia.org/wiki/Danish_and_Norwegian_alphabet). I just wanted to point out that the community overwhelmingly cites this paper with the author's name using a slashed O. And, subsequent work by Gunnar Ostrem uses the slashed O.

Figure 1: I would consider giving all these acronyms in the caption. Also, the second to last step isn't quite clear - what is "both" here?

Figure 2: This figure made me look back to the text to make sure if I understood that the maximum debris extent would merge the evolution of the event. I think that is true, regardless of the search timeframe (and somewhat dependent on the Landsat image separation). This would mean that to track the debris transport you would first find that an event occurred and then go back and look through all images to characterize how

it evolved - this is all a user step, right? I'm thinking of your Lituya Mountain example: if you instead ran GERALDINE for the timeframe of 2012-2014 you would get one maximum extent and then you would have to notice that the event occurred in 2012 and was still visible in 2013 and 2014 frames. This is still great since it is relatively little work to analyze around a particular event compared to finding the event in the first place. Right? I think some more context on how many events may exist and how laborious it is search individual frames may help put this in context. And, you could say a bit more about this workflow in Section 3.3, since what is said around line 260 isn't quite clear how that connects to what is shown in Figure 2 (if at all).

Supplement:

- Section 4.0 first paragraph should be "complementary" instead of "complimentary"

---

## Referee Comment (RC3) · Sam Herreid (Referee) · 24 Jul 2020

The article "GERALDINE (Google earth Engine supRaglAciaL Debris Input dEtector) – A new tool for Identifying and Monitoring Supraglacial Landslide Inputs" By Smith at al. describes a tool that subtracts composite debris maps from two stacks of Landsat images, one from a period of interest and the other from the preceding year, to isolate new debris additions. A user can then interpret this output to locate supraglacial rock avalanches or landslides. I found the paper mostly easy to read and I think the research objective is timely and useful. I also appreciated the user guide provided in

the supporting information. However, I think the authors stopped their tool development prematurely leaving some fundamental elements unaddressed. My main points of concern are briefly summarized here with more detailed comments inline below along with minor comments.

Looking at the two map figures of the article, it is clear that, even within the GEE stack methodology, which is in principal sound, debris cover is not confidently mapped. There is unphysical debris in the accumulation zone and many instances of "new debris additions" that are not new debris additions. These areas accumulate into tool output false positives that are neglected by the authors who rather only report true positive success, leading to statements like L283-284: "GERALDINE outputs [had a] 100% successful identification". By neglecting to calculate a metric like precision or the false positive rate, the study is lacking a meaningful assessment of performance. I think it is reasonable to state, as the authors do, that some of these debris map errors stem from errors in the RGI, but these then need to be either mitigated or quantified in the error assessment of your tool.

From my view, the main incentive for a tool that considers every image acquired in a stack, is to detect rock avalanches that are deposited onto a glacier's accumulation zone and automatically assign a best constrained deposition date. The automated detection of rock avalanches deposited onto bare glacier ice in ablation zones is also useful, but there is less chance of missing one since there will be a surface expression in every snow/cloud free image after deposition until it is too heavily reworked or evacuated from the glacier. Further, in ablation zones there is the case, that will likely only grow in frequency, where a rock avalanche is deposited onto existing debris cover, or earlier deposited rock avalanche debris, which is an entirely undetectable event using this method. By summing debris cover over one year or longer, the method presented here will likely catch a deposit onto the accumulation zone, but by not finding the difference between each sequential image the approach loses any ability to assign a deposition date. I understand the incentive to aggregate debris, but from the comment above, I think the quality of the resulting debris maps are still low relative to other automated debris maps in the literature.

I think a rock avalanche deposit onto bare glacier ice is a strong signal that can be detected automatically. For example, the area of a rock avalanche feature will almost always be much larger than any other location of debris additions from other sources (if mapped accurately and dt is short, e.g. 1 year). The authors leave this step to the user which I think significantly reduces the applications of this tool. I can accept that this version does not need to perfectly resolve all of elements to mapping rock avalanches onto clean glacier ice, but I think providing an automated selection of rock avalanches from new debris additions is only a minor addition that will increase both the tool application as well as ability to quantify true positives, false positives and false negatives. I also think that looking at the differences between every image after a rock avalanche is detected to constrain the date of deposition is a reasonable and achievable result at this stage of tool development.

If this method is to be a starting point for a globally applicable tool (L22), I am concerned that the authors cite limitations of GEE that cause the region of interest to be limited to <5000 km2. Do the authors anticipate that this method could be written in a more computationally efficient way such that this limit will be dramatically increased? Highly useful functionality of a tool like this one will be when all of Earth's glaciers can be assessed in near real time, but if there are intrinsic limitations within GEE is this a feasible future for this tool?

Finally, there is a factor present in the quantity "new debris additions" that is not quantified or discussed. Unstable glacier flow will produce debris structures that deviate from flow lines parallel to a glacier's valley wall (e.g. the surge loops on Susitna Glacier in your Figure 4) and a difference map of debris cover over some dt will show a false gain and false loss of debris cover that is really just debris structure translation. Where glacier flow instabilities are present, a simple difference of debris cover maps cannot be strictly new debris additions. Herreid and Truffer, 2016 provides a discussion on this

topic.

Herreid, Sam, and Martin Truffer. "Automated detection of unstable glacier flow and a spectrum of speedup behavior in the Alaska Range." Journal of Geophysical Research: Earth Surface 121.1 (2016): 64-81.

L1: Perhaps stylistic but I think "A new tool for identifying and monitoring supraglacial landslide inputs" is a better title, without the less straightforward and somewhat redundant acronym.

L9: Why not use "rock avalanche" throughout? I believe rock avalanche is more precise and consistent with the literature for what you are looking at. If the authors prefer the more general term landslide, then early in the introduction make clear what is and is not a landslide vs rock avalanche for this study and keep the language consistent. It's strange to read landslide in the title and have rock avalanche be the first sentence of the abstract.

L9-12: There is a missing step here, rock avalanches can happen far from glacier ice. Detection of RAs for the study of RAs alone, or to answer frequency questions with respect to climate or ice factors, should consider all RAs independent of their runout happening to be on a glacier. This is either a very big sampling bias or you should pose a glacier specific problem.

L14: It reads like you are focusing on filling this small to medium gap but on L43 you say you focus on the inputs of high magnitude, $> 10^6$ m$^3$, RAs. Please clarify/fix and keep consistent throughout. L215 considers a 0.062 km2 event.

L22: From the abstract alone you don't mention measuring area or volume or event timing, so I don't quite see the jump to a global product. Further, on L118 you advise ROIs <5000 km2. Do you anticipate a less computationally costly version of your method or are there HPC options in GEE? Finally, it is a little strange to have a first step towards a revision, a revision implies several steps have already been taken.

L26: With the known errors in the RGI, it's better to avoid presenting the number of glaciers to the accuracy of a single glacier. Consider ">200,000".

L27: Consider a revised global estimate of debris cover from Herreid and Pellicciotti, accepted by Nature Geoscience, which should be available by August 2020 at this DOI: 10.1038/s41561-020-0615-0

L34: Either add "e.g." to the citations or also add a citation to Kirkbride and Deline, 2013 whose Table 1 gives a more complete list of citations for expanding debris cover.

Kirkbride, Martin P., and Philip Deline. "The formation of supraglacial debris covers by primary dispersal from transverse englacial debris bands." Earth Surface Processes and Landforms 38.15 (2013): 1779-1792.

L35: What is the difference between sub- and en- glacial sediments in this context? I don't think sub-glacial sediments can melt out.

L35: Anderson, 2000 addresses general dispersion of medial moraines which you don't explicitly mention here. Does "debris store" mean extraglacial debris? This is not clear.

Anderson, Robert S. "A model of ablation-dominated medial moraines and the generation of debris-mantled glacier snouts." Journal of Glaciology 46.154 (2000): 459-469.

L36: It might be worth distinguishing here high volume low frequency mass movements from low volume high frequency.

L43: How are you able to focus on landslide of a particular volume? Throughout you do not calculate or consider volumes. And do you mean high volume? Magnitude of what?

L44: "where there is disparity between current high rates of activity above ice" this is unclear.

L46: lag ice-free conditions in terms of what?

[Figure]

L47 What does "relatively low in the landscape" mean?

L58: I'm not sure if there are remote sensing methods yet to see englacial debris. Maybe you mean geophysical methods, e.g. GPR.

L59: "[add: potentially] considerable modification"

L60: "Deposited"? "Emplaced" is odd.

L72: Landslides vs RA confusion here.

L87: Open access or open source?

L90: Define what you mean by "wide" in parentheses

L109: RGI errors are further quantified in Herreid and Pellicciotti, accepted by Nature Geoscience, available around August 2020 at DOI: 10.1038/s41561-020-0615-0

L116: add: "[and all images in the] year preceding..."

L118: What do you mean by "specify annual date ranges"? Are you saying the tool can only work for one time window between two specified years? This seems like a pretty critical limitation to the functionality to the tool. Are you sure GEE is the correct platform if its memory capacity is such a bottleneck? Maybe JuypterLab is a better cloud-based platform? Or your code could select a single optimal image of a one year stack and then make your calculations on single images? Also if you clip the RGI first, then all of your calculations will be less computationally costly.

L122: This section is not very clear, but if I understand correctly, the tool will collect two stacks, one from the year before a defined date range and one for the full defined date range, and then perform a single subtraction to find a single map of new debris. There is an issue of accumulating "new debris additions" if the stack of images aggregate debris from, say, 10 years, there will be much more new debris additions that are not sourced from RAs. You also lose the ability to automatically detect a deposition date which is, in my view, the main incentive to use GEE and consider stacks of images

rather than single optimal images. I think maybe you should change the wording of a "user-specified date range", and rather say "a user specified year where the tool will give you a map you can look for RAs deposited since the preceding year." But 1. I don't understand why finding the RAs can't also be automated, this should be a very clear signal if deposited on clean ice (you will entirely miss RAs that are deposited onto existing debris cover); and 2. As a user I can think of two uses for a tool like this: (a) getting the location of all RAs that have been deposited onto a glacier and are still present at the surface and a deposition date if deposited since Landsat 4; and (b) near-real-time detection. I think your tool could be successful for the latter, although to be practical it should be able to analyze all of Earth's glaciers at once or at least all glaciers in, say, Alaska (Bearing Glacier in SE Alaska alone is larger than the recommended <5000 km2 ROI), but I think there is still a lot of improvement needed for the former. The difference map needs to be computed annually to keep other debris addition signals small and also facilitate a deposition date.

131: I can appreciate that the method used to assess cloud mask performance considers clouds in an entire stack, thus incorporating a variety of cloud types in a simple run of your code. However, I would like to see more direct evidence that clouds themselves are accurately mapped. From my experience cloud mapping algorithms are unreliable in glacierized areas. Could you show a side by side image of a raw satellite image and an overlay of the output of the cloud mask with scores >20%, perhaps one where it worked well and a second where it was at its worst. I'm concerned that you're only mapping 60% of RA area. How were the studies that make up your validation dataset able to map 100% of the RA area and you cannot? Surely with the stack methodology the aggregate over many images should, together, capture 100% of RA area unless it's a particularly snowy or cloudy year. Does this suggest you have a 40% error rate in detecting RAs?

L132: What about cast shadows from topography? Herreid and Pellicciotti, accepted by Nature Geoscience (available August 2020 at DOI: 10.1038/s41561-020-0615-0)

found it necessary to remove area in shadow in order to accurately map debris cover. The band ratio method is able to negotiate some shading, but when a surface becomes too dark there is still the possibility for false positive debris classification (e.g. Herreid and Pellicciotti, 2020 removed 760 km2 of shaded glacier area in Alaska and Western Canada).

L134: I don't really see a justification for the step of mapping supraglacial lakes or ponds. These features generally develop in heavily debris-covered portions of glaciers where your tool will fail to detect a RA by not having the prior bare ice context. Further, if these features are 22 pixels on average, as you cite in the SI, then the above discussed 40% omission error dwarfs the stream/pond signal. If you elect to keep this component please provide an example in the SI that shows how mapping streams and ponds leads to a higher rate of RA detection.

L150: One of your inequality signs should include "or equal to"

L158: There is a missing discussion on double counting translated debris features that deviate from a flowline parallel the glacier valley walls. Also summed non-RA debris additions if the user defined time period is not sufficiently short. Herreid and Truffer, 2016 established a very similar methodology to the one presented here in order to detect glacier flow instabilities. In this study RA are identified but considered an error in the context of the flow instability research question. For your work, RAs are signal and the features identified by Herreid and Truffer, 2016 are errors. These should be discussed.

Herreid, Sam, and Martin Truffer. "Automated detection of unstable glacier flow and a spectrum of speedup behavior in the Alaska Range." Journal of Geophysical Research: Earth Surface 121.1 (2016): 64-81.

L162: What do you mean by "Debris biased"?

L168: Do you mean an omission/commission validation? If not, please provide an

additional sentence on why a bipartite approach was used.

L172: RA already defined.

L175: 48 suitable events were found out of how many that you considered? It is helpful for the reader to know if these are rare occurrences or the majority. I assume these inventories only consider supraglacial RAs?

L175: please add a map figure showing all of the regions you applied your tool

L189: I think if your code mapped RAs from the best available image for each event, rather than a composite, you could be very close to 100%.

L189: A relevant factor that you do not mention is a RA that crosses existing debris cover. This is likely the predominant factor of why you will not be able to map RAs to 100%.

L196: The accuracy of the satellite image remains the same, the overall significance of a single pixel of a small glacier increases.

L197: Looking at the noise in bare ice regions of Figure 4 I struggle to see what you mean by "true negative detection rate is also extremely high"

L198: I don't agree with this justification for user verification. If you subtract two optimal satellite images before and after a RA deposition onto a non-debris-covered portion of a glacier, the signal is exceptionally prominent, and I see no reason why an algorithm cannot easily identify this automatically. I think somewhere in your GEE stack processing, the debris mapping and the cloud removal, a very clear signal becomes muddy. I think some small changes to your workflow can provide a much clearer, and likely more computationally efficient output.

L199: The problem with saying "to a user familiar with glacial and landslide processes, the [tool output is] clear" is that a user familiar with glacial and landslide processes will be able to spot large landslides onto bare ice from a raw image. The spatial domain

of the tool is low <5000km2 and the tool cannot iterate over many years to pinpoint a deposition date. I think there is a lot of potential in a tool like this but in its current state I have a hard time seeing a scientific application.

L202: Please add a section to methods describing how your derived areal extent. Presumably there was a manual step involved in this.

L215: How much user interpretation was involved with isolating the 71% true-positive RA area? False positive and false negative areas must also be considered to make a statement about detection confidence.

L228: But if topographic shading is classified as debris, it will influence new debris detection.

L247: Your method has a high potential to detect all events [add: that are deposited onto initially bare glacier ice]. E.g. a hypothetical second event at the same scarp on the glacier east of Maclaren Glacier that deposited a slightly smaller volume of rock would be entirely missed by your method.

L250-256: I find this to be significant conditionality and required prior knowledge for an automated tool. Your method doesn't automatically run for multiple years sequentially, so how would someone new to the area know where to start? Reading your Figure 4 alone suggests the BRG RAs were deposited between 2017 and 2018, this is misleading. The mapped Lituya RA in Fig. 5 also appears patchy, should the logic of L254 be followed and this area be dismissed as erroneous?

L257: While translated features are present in your output (also translated features from flow instabilities, see Herreid and Truffer, 2016) and are scientifically useful, these are errors with respect to your intended tool function. If you can automatically differentiate feature translation from feature deposition then this can be a nice side component to your study, otherwise I think you need to treat this as error.

Herreid, Sam, and Martin Truffer. "Automated detection of unstable glacier flow and a

spectrum of speedup behavior in the Alaska Range." Journal of Geophysical Research: Earth Surface 121.1 (2016): 64-81.

L275: How does reduced ablation over one year around the ELA, where ablation rates are generally low, increase surface velocities?

L281: RAs on bare glacier ice in ablation zones are easy to identify from one recent image and your method also requires manual inspection. Here I think you should highlight your tool's ability to potentially catch events in the accumulation zone that have only a very short residence time.

L284: This is the first mention of 100% successful identification which should first appear in the results section, but I also think it is incorrect. By considering only true positive area, a map that is entirely "new debris additions" will also have a 100% successful identification rate but is clearly meaningless. You need to score your success against false positive and false negative area.

---

## Author Comment (AC1) · 17 Sep 2020

**Response to Reviews for: "GERALDINE (Google earth Engine supRaglAciaL Debris INput dEtector) – A new Tool for Identifying and Monitoring Supraglacial Landslide Inputs" (MS No: esurf-2020-40)**

Dear Dr. Conway,

We thank you and the reviewers for your time, and constructive and helpful comments on our manuscript. We have addressed each one of the comments below, and, have added suggested manuscript additions formatted as "new text" where appropriate.

Reviewer comments are *grey highlighted*, and, use their numbering where it was used in the review.

William Smith, on behalf of all authors.

**Gioachino Roberti (Reviewer 1)**

*I enjoyed reading the paper "GERALDINE (Google earth Engine supRaglAciaL Debris Input dEtector) - A new Tool for Identifying and Monitoring Supraglacial Landslide Inputs" and I recommend it for publication. The paper presents a new tool to exploit Landsat images in Google Earth Engine to map debris onto glaciers, therefore providing a semi-automatic tool to identify rock avalanches emplaced on to glaciers, and to track supraglacial debris movement. This tool can complement seismic analysis, and, if extensively applied, help developing F-M curves of rock avalanches onto glaciers in the past 37 years.*

We thank the reviewer for their time and comments. We have addressed each one of these below.

*The following comments can help to further improve the paper. I think a better overview of satellite spatial resolution and detectable landslide size is needed in introduction.*

We agree and will add a new paragraph on line 87 as follows:

"Since the launch of Landsat 1 in July 1972, optical satellites have imaged the earth surface at increasing temporal and spatial frequency. Six successful Landsat missions have followed Landsat 1, making it the longest continuous optical imagery data series, revolutionising global land monitoring (Wulder et al., 2019). Analysis ready Landsat data is available for Landsat 4 (1982-1993), Landsat 5 (1984-2012), Landsat 7 (1999-present) and Landsat 8 (2013-present), providing 38 years of data at a 30 m spatial resolution and a 16-day temporal resolution. These data are categorised into three tiers: (1) Tier 1 data that is radiometrically and geometrically corrected (< 12 m root mean square error); (2) Tier 2 data which is of lower geodetic accuracy (> 12 m root mean square error); and (3) Real Time imagery, which is available immediately after capture but uses preliminary geolocation data and thermal bands require additional processing, before being moved to its final imagery tier (1 or 2) within 26 days for Landsat 7, and 16 days for Landsat 8. Traditionally, it has been difficult to exploit extensive optical imagery collections such as Landsat, without vast amounts of computing resources. However, in the last decade, cloud computing has become increasingly accessible. This allows a user to manipulate and process data on remote servers, removing the need for a high-performance personal computer. Google Earth Engine (GEE) is a cloud platform created specifically to aid the analysis of planetary-scale geospatial datasets such as Landsat and is freely available for research and education purposes (Gorelick et al., 2017).

Here, we utilise Google Earth Engine (GEE), and the Landsat data archive of 37 years of optical imagery, to present the Google earth Engine supRaglAciaL Debris INput dEtector (GERALDINE). An open-source tool to automatically delimit new supraglacial landslide deposits over wide areas and
timescales…"

We shall remove any information from the methods section that is explained in this paragraph.

Wulder, M.A. et al.: Current status of Landsat program, science and applications, Remote Sensing of
Environment, 225, 127-147, doi: 10.1016/j.rse.2019.02.015, 2019.

*In the discussion section you could add some paragraphs:*

*1) A paragraph about development F-M curves, as it is a topic mentioned in introduction and conclusion*
*but not directly addressed in the discussion*

We do not present any results in this primarily methodological paper that revise magnitude frequency
relationships. We consider this outside of the present scope of the paper, which is to present the tool,
its capability, and to validate it against known events so it can be applied to specific areas in the future.
However, it is an important point for the introduction and discussion as this is how the tool will be
applied in future (by us, and, by others) once the tool is accepted and published. In this study we are
signposting the user to use GERALDINE with confidence, even in areas with existing landslide
inventories where GERALDINE can/should be tested to confirm or modify F-M curves. We will revise
the introduction to include more detailed information of recently published inventories of glacial RAs,
which provide a basis for F-M testing. On line 71 we will reword and add:

"Manual imagery analysis to identify supraglacial landslide deposits and RAs has principally been
applied in Alaska. This technique enabled detection of 123 supraglacial landslide deposits in the
Chugach Mountains (Uhlmann et al. 2013), 24 RAs in Glacier Bay National Park (Coe et al. 2018), and
more recently, 220 RAs in the St Elias Mountains (Bessette-Kirton and Coe, 2020). These studies
acknowledge that their inventories are incomplete/underestimates due to analysis of summer only
imagery and an inability to detect events that are rapidly advected into the ice. These are critical
drawbacks preventing accurate magnitude frequency relationships from being derived but analysis of
more imagery over larger areas is unfeasible due to time and computational requirements. Studies of
this kind are also typically in response to a trigger event e.g. earthquake or a cluster of large RA events
(e.g. Coe et al. (2018) in Glacier Bay National Park), spatially biasing inventories into areas with known
activity. They therefore provide a snapshot in time, with no continuous record. Methods are needed
which are accessible, quick and easy to apply and require no specialist knowledge, to re-evaluate
magnitude frequencies in glacial environments. Currently, the only method capable of identifying a
continuous record of such events, is seismic monitoring (Ekström and Stark, 2013). Seismic detection
utilises the global seismic network to detect long-period surface waves, characteristic of seismogenic
landslides. Seismic methods have identified some of the largest supraglacially deposited RAs…"

Bessette-Kirton, E.K. and Coe, J.A. A 36-Year Record of Rock Avalanches in the Saint Elias Mountains
of Alaska, With Implications for Future Hazards, Frontiers in Earth Science, 8:293, doi:
10.3389/feart.2020.00293, 2020.

*2) A paragraph about "eliminating the time-intensive process of manually downloading, processing*
*and inspecting numerous satellite images" that is then mentioned in the conclusion. With considerations*
*about transferability of the method to other satellites and data storage and processing platforms*

We fully agree this needs quantifying given it is a major benefit and will add this brief section to line
233:

"GERALDINE used a total of 228 Landsat images for analysis; 107 to determine the 2017 debris extent
and 121 to determine the 2018 debris extent. Landsat tiles vary from 200 MB to 1000 MB  when
compressed, so, if we assume an average tile is 500 MB, a user would require 114 GB of local storage,
a large bandwidth internet connection to download (which comes with an associated carbon cost), and, a PC capable of processing these data. GEE required none of these requirements and completed analysis
in under two minutes, extracting information from every available cloud-free pixel, to maximise use of
the imagery. The new debris output map produced was 6.5 MB, and contained all relevant 'new' debris
information from 2018."

*3) and (eventually) how a similar approach may be used in other context (landslides in forested areas*
*etc.)*

We understand why the reviewer asks this question, but the methodology used as presented here is not
suitable for use in other environments and has been deliberately tuned/developed to detect in snow and
ice landscapes where landslide deposits have little residence time. The ability to threshold a band ratio
technique into two distinct categories (ice/snow and debris), is what GERALDINE exploits. In contrast,
there are multiple substrates which must be categorised in other environments, and therefore a different
method is required. Scheip and Wegman exploit percentage change and NDVI in their tool (developed
at the same time as ours) which is tuned to vegetated landscapes. We will make reference to:

Scheip, C.M. and Wegmann, K.W. HazMapper: A global open-source natural hazard mapping
application in Google Earth Engine, Nat. Hazards Earth Syst. Sci. Discuss., doi: 10.5194/nhess-2020-
108, in review, 2020.

**In line comments:**

*Line 14: Quantify? What's the size of the smallest detectable landslide?*

A specific size is not possible because it depends on the strength of the seismic signal. We shall clarify
and reword to:

"Although large landslides can be detected and located using their seismic signature, smaller landslides
($M \leq 5.0$) frequently go undetected because their seismic signature is less than the noise floor,
particularly supraglacially deposited landslides which feature a "quiet" runout over snow."

*Line 18: You can detect only the large landslides? This sounds in contrast with your earlier statement*

We shall amend to "from which large debris inputs such as supraglacial landslide deposits ($> 0.05$ km$^2$)
can be rapidly identified"

*Line 21: Ok cool. So large landslides that may not have been identified seismically. I don't think you*
*need to try to "sell it" as alternative or better method than seismic identification, they can be used*
*together. A tool like the one you have developed is cool by itself, seismic identification or not.*

We think that highlighting that GERALDINE can detect landslides, which are both seismically and
non-seismically detectable is important in the abstract as it is not a well-known point. However, we do
explain in the manuscript, that the available tools should be used in conjunction – this is now happening
with the authors of the original seismic landslide detection paper.

*Line 24: Very cool! But you should expand the 37 year F-M topic in the discussions*

See above comments (Line 47-77 of our response) on F-M topic. We think this is the end-point use of
GERALDINE, establish a past supraglacial inventory globally, then run this as near to live as Landsat
allows. However, we wish to have GERALDINE published and accepted, and in use by others, as we
continue to try and develop/collate F-M for regions in collaborative ways.

*Line 38: Can you method distinguish these 3 types of debris cover?*

GERALDINE cannot distinguish between debris types. We think it is important to highlight all ways
of debris transport into supraglacial environments, so the user has an idea of all debris sources/pathways
and can evaluate fluxes.

 *Line 40: How do you distinguish re-emerged debris vs supraglacially emplaced debris?*

It is not possible to use GERALDINE to identify the different debris transport pathways from remotely
sensed data alone and would require further analysis in the field.

*Line 43: Ok, you should be clearer about the landslide size in the abstract too*

We are going to remove the landslide size from this section of the work as GERALDINE can detect
events smaller than this. We will reword to:

"Here we focus on supraglacial landslide deposits (>0.05 km$^2$), commonly associated with RAs, defined
as landslides: (a) of high magnitude (> $10^6$ m$^3$); (b) perceived low frequency; (c) long runout; and (d)
where there is disparity between high present-day rates of slope processes above ice (Allen et al., 2011;
Coe et al., 2018) and expected rates based on theories of lagged paraglacial slope responses (Ballantyne,
2002; Ballantyne et al., 2014a)."

*Line 80: How small is "smaller landslides"*

In the context of seismic detection, this is difficult to define because it depends on the seismic signal,
which can be determined by a plethora of things such as: source area, volume, drop height and horizontal
runout distance. We think it is better to state the magnitude threshold from which they are difficult to
determine but we will make this clearer by rewording Line 80 to:

"This also results in an inability to detect landslides that are relatively low in volume, due to their weak
seismic fingerprint ($M \leq 5.0$)…"

*Line 82: It sounds like you are detecting large landslides that have no seismic signature rather than*
*"small landslides"... maybe you can reword a bit to put emphasis on the combination of size, frictional*
*melting etc.*

This sentence is simply to describe the limitations of the seismic method for detecting landslides onto
glaciers. We shall reword Line 82 to emphasise that these are two substantial inter-related drawbacks
of the method:

"This also results in an inability to detect landslides that are relatively low in volume, due to their weak
seismic fingerprint ($M \leq 5.0$) and causes underestimation of landslide properties (e.g. event size and
duration) because their runouts are seismically "quiet", likely due to frictional melting of glacier ice
(Ekström and Stark, 2013)."

*Line 89: It may be worth expanding this paragraph/add new paragraph and give an overview of GEE*
*and Landsat satellites. I see you discuss landsat satellites in method and validation sections but an*
*overview of the satellites (different tiers, spatial resolution, accuracy, years of operation and revisiting*
*time etc...) in the intro will help the reader.*

As discussed on line 20-45 of our responses, we will add a new paragraph discussing Landsat and GEE.

*Line 97: I think the resolution should be mention earlier in the paper too*

Agreed. See above additional paragraph (line 20-45 of our response) in response to line 89.

*Line 197: Would this still be useful to assess some of the other supraglacial debris types presented in*
*the introduction? Expand on this (see my comment of figure 5).*

As mentioned above it is not possible to distinguish between debris types, but on line 277 we will
discuss different types of debris and add:

"We note other areas are flagged as 'new debris' in 2013 and 2014. These are typically where glacier
downwasting has occurred exposing more of the valley walls, or where there has been temporal evolution of the debris cover e.g. glacier flowline instabilities. These flow instabilities can cause double-
counting of debris when larger time windows are specified (see Herreid and Truffer, 2015). Both
processes subsequently cause false classification as 'new debris'. However, neither glacier
downwasting nor evolution of the debris cover display supraglacial landslide characteristics, so it is
highly unlikely that a user would mistake them for one."

Herreid, S. and Truffer, M. Automated detection of unstable glacier flow and a spectrum of speedup
behaviour in the Alaska Range, Journal of Geophysical Research: Earth Surface, 121(1), 64-81, doi:
10.1002/2015JF003502, 2016.

*Line 217: maybe you could also briefly discuss how this method could be applied (maybe not in GEE*
*but in some other environment) to other satellites*

We shall add this to the end of the paragraph (Line 219):

"We also envisage development with other higher resolution and higher repeat satellites e.g. the Sentinel
2 and Planet Lab constellations. However, we found that current cloud mask algorithms for these data
are not sufficient for accurate global glacial debris delineation."

*Line 276: In the intro you mentioned the possibility of the development of frequency/magnitude curves*
*for landslides onto glaciers, but there is not discussion of that point here. Maybe you can add a short*
*paragraph (with example?) to explore that potential.*

We hope the above comments (Line 47-77) on M-F topic resolve this.

*Line 277 Something else that feels like may be missing in the discussion is the overview of the value of*
*"eliminating the time-intensive process of manually downloading, processing and inspecting numerous*
*satellite images" that is then mentioned in the conclusion*

See above (Line 77-89 of our response) for section which will be added to discuss this to Line 233.

*Line 287: I agree, but you should discuss this in the discussion. See my comment about frequency-*
*magnitude*

See above comments (Line 47-77) on F-M topic.

*Figure 4: Can you can mark the collapse scar of these landslides? will help the reader*

Agreed. We will add a source scar.

*Figure 5: Same, where is the landslide coming from? Can you discuss the origin of the other debris*
*addition in the glacier on the top right of the picture? in relation to my comment at Line 197.*

We will highlight the source scar in the image. See above comments (line 162-176 of our response)
regarding Line 197.

**Michelle Koutnik (Reviewer 2)**

*In this study the authors develop a powerful new tool to identify supraglacial landslides. They present the tool, as well as demonstrate how it can work and the value of it by identifying two previously unknown landslide debris events. This is an exciting development, and valuable to capturing these events where evidence of them is often lost quickly on the landscape, and yet they are important debris sources. I enjoyed reading the paper and I really enjoyed thinking about what may be possible using the tool. I have some questions and suggestions for the authors, but all of these points are minor.*

*Overall, great work on this*

We thank the reviewer for their time and comments. We have addressed each one of these below.

*1) It could be worthwhile to put the size of landslide deposits that you can identify in more context with the size of glaciers that you can reliably search over and/or something about the size distribution of glaciers around the world. You don't have to answer this but it made me curious: over what proportion of the total number of glaciers would be possible to detect a rock avalanche of the size that you search, assuming that an event occurred? Is this the same as the base number of glaciers with debris mentioned, which was 4.4% of 215,547 glaciers worldwide? And, the abstract mentions >2km2 area but around line 43 the mention is volume. It would be helpful to relate these together and also indicate how volumes are estimated. With respect to the events it may also help to explain why these are referred to as 'high magnitude' - is this your designation?*

We thank the reviewer for raising an interesting point. We will reword line 26 to:

"There are currently >200,000 glaciers worldwide covering >700,000 km$^2$, of which 8.2% are less than 1 km$^2$ (Herreid and Pellicciotti, 2020), excluding the Greenland and Antarctic ice sheets (RGI Consortium, 2017)."

The abstract states that for the area of the deposits we show examples of, our actual minimum deposit size that we have confidence in is 0.05 km$^2$. This is mentioned on Line 216. We will omit the volume from line 43 and utilise "supraglacial landslide deposit" as an umbrella term for all events as volume requires an (not well agreed on) empirical relationship to be applied. We will reword this section to:

"Here we focus on supraglacial landslide deposits (>0.05 km$^2$), commonly associated with RAs, defined as landslides: (a) of high magnitude ($> 10^6$ m$^3$); (b) perceived low frequency; (c) long runout; and (d) where there is disparity between high present-day rates of slope processes above ice (Allen et al., 2011; Coe et al., 2018) and expected rates based on theories of lagged paraglacial slope responses (Ballantyne, 2002; Ballantyne et al., 2014a)."

Herreid, S. and Pellicciotti, F. The state of rock debris covering Earths glaciers, Nature Geoscience, doi: 10.1038/s41561-020-0615-0, 2020.

*2) Another question is if this tool could detect smaller-scale events. Is it that any smaller events are not considered rock avalanches and/or that they cannot be detected? (I thought that "rock avalanches" were defined being >1Mm3, but I could be wrong about that) What about rock avalanche events on already heavily debris-covered glacier surfaces - would those be detectable?*

As above (Point 1, line 201), we are going to be much clearer about this in the text and reword sections of the manuscript referring to all large debris inputs detected (>0.05 km$^2$) as "supraglacial landslide deposits". This is because we do not know the processes which resulted in slope failure, and, as you rightly point out changing between different terms is confusing. We believe that one umbrella term such as "supraglacial landslide deposit" will address this. However, we do validate GERALDINE against RA deposits as this was the primary design for the tool and there are good inventories to test against, as these examples have been investigated, confirming their failure/deposition process.

We shall add a sentence at L228 addressing multiple failures:

"GERALDINE can also not detect landslide debris deposition onto an existing debris cover. Therefore,
if a supraglacial landslide consists of multiple failures, a GERALDINE output map would only detect
one event, with the deposit extent being the combined total of all failures. It would be highly beneficial
to combine GERALDINE with seismic detection to help delineate the amount of failures that occur."

*3) It could also be worthwhile putting the need / value of this tool in context with the total number of*
*rock avalanches of this scale that have been found to date. Was the validation set of 48 known events*
*chosen to span as many regions as possible, or are these all of the events that have been catalogued to*
*date? More context on the likelihood to find additional, unidentified events would be helpful. Another*
*way to expand on that could be to illustrate just how labor intensive it would be to search the Landsat*
*archive manually. What is the range of repeat times of Landsat? This would also help put in context the*
*two new events that you did identify.*

We provide reasons why those 48 validation RAs were suitable for this study on Line 173 and will add
a map of their locations in the supplementary information. The cited sources feature the largest datasets
of supraglacial RAs, which does induce some spatial bias but they do span a range of conditions and
glacial landscapes. We will add more recent context (RA frequencies in Alaska) in the paragraph
beginning on Line 68 (see lines 47-77 of Gioachino Roberti's review).

As addressed in Gioachino Roberti's review (reviewer 1), we will add a section in the introduction (see
lines 20-45 of our response to Gioachino's Roberti's review) detailing the Landsat data archive
(resolution, repeat times and data tiers) and in section 3.2 we will give an example of the local
requirements needed to undertake the Hayes range identification of both RAs (see lines 77-89 of our
response to Gioachino Roberti's review).

Finding additional, unidentified deposits is one of the driving purposes of the tool, both in areas already
studied, and in those with no inventory. This work is underway, but, relies on GERALDINE being
accepted by peer review as being able to reliably identify supraglacial landslides, based on known
validation events. The revision of inventories and new inventories is the logical following paper.

*Related to this point: I may have missed it, but how computationally and user-labor intensive is applying*
*this tool. It sounds well beyond the scope of what a team like yours could do, but how far from possible*
*would it be to search all glaciers where events may have occurred for the past 37 years? Is the challenge*
*on the GEE computation side or on the validation side? When the latest RGI outlines come out is this*
*something that could be done?*

As addressed in Gioachino Roberti's review (reviewer 1) (see lines 77-89 of our responses), in section
3.2 we will explain the amount of images GERALDINE processes for the Hayes region in 2018 and the
computational storage/processing/time that it saves. The challenge would certainly be validation of
events over very large spatial extents (necessary to ensure accuracy), but it would be possible if
numerous people worked on different regions and would be an interesting avenue for future work.

*It may be your goal to let the curiosity of the users take over here, but are there some outcomes you*
*think this makes possible in the short term and would advocate for (or may be doing yourselves?). Not*
*having a sense of how intensive the process is, I was left to wonder the scope of study that may be*
*reasonable to undertake - maybe in the conclusions you could indicate something about studies that*
*seem worthwhile? For example, is this best applied to target regions of a certain size and/ or over target*
*timeframes of a certain duration?*

We will highlight and cite a recently published paper, which calls for systematic, long-term observations
of RAs and the regions suggested for analysis (Bellwether sites). On line 86 we will add:

"These links, coupled with the availability of high spatial and temporal resolution optical satellite
imagery, have demonstrated the need for systematic observations of landslides in mountainous
cryospheric environments (Coe, 2020). Five 'bellwether' sites have been suggested for these purposes:
the Northern Patagonia Ice Field, Western European Alps, Eastern Karakorum in the Himalayas,
Southern Alps of New Zealand and the Fairweather Range in Alaska (Coe, 2020).

We think the focus should be on six main areas, which we will highlight by adding to Line 287:

"We suggest users should apply GERALDINE at standardised time intervals in recently identified
'bellwether sites' (Coe, 2020) in glaciated high mountain areas undergoing rapid change i.e. Greenland,
Alaska, Patagonia, the European Alps, New Zealand Alps and the Himalaya, to investigate annual rates
of these large debris inputs."

Coe, J.A. Bellwether sites for evaluating changes in landslide frequency and magnitude in cryospheric
mountainous terrain: a call for systematic, long-term observations to decipher the impact of climate
change, Landslides, doi:10.1007/s10346-020-01462-y, 2020.

*In the supplement it was mentioned that you compared to a Planet image. Is there anything that can be*
*said about the future of applying this tool to other image sets? I understand that Planet images are not*
*openly available, but is Landsat the only archive that makes sense to use? Is there anything to say about*
*coordinating Landsat-based results with other image sets, or does that just need to be taken on a*
*glacier-by-glacier basis? This would be important to at least mention, but doesn't take away from the*
*achievement of getting this to work for Landsat data.*

This was also raised by Gioachino Roberti's review (reviewer 1) and is a good point. We will add a
paragraph to the introduction explaining that the time-span of Landsat data makes it most suitable for
this purpose, and that Sentinel/Planet imagery incorporation is a future goal but that cloud masks for
these datasets are currently too inaccurate (see line 177-182 of our response).

*4) This is a subtle point, but it seemed like one that was done deliberately in the text so wanted to raise*
*my reaction. The title (and acronym for the tool) uses "identifying" to describe what is done by the tool.*
*And, the tool is referred to as a "detector". However, typically the text refers to what the tool provides*
*as "highlighting" new events. It is only after user evaluation that they are "identified". If this was*
*deliberate then I would check for complete consistency and maybe say that directly somewhere. I*
*suggest that identify (or detect) is a reasonable term for what the tool does, and then the user confirms*
*or validates that finding - but, any word choices you prefer will work as long as explained clearly and*
*used consistently. (Pay particular attention to this in the conclusion where the language seems to be*
*mixed.)*

We thank the reviewer for raising this important point and shall check for consistency of these terms.
We agree that the tool is a detector to aid identification of large debris inputs, but we also believe that
saying tool outputs 'highlight' these events is also in-keeping with the language.

*Another subtle point on language is if all "supraglacial landslide inputs" are the same as "rock*
*avalanches"? And, assuming that debris inputs are also the same thing? I would be check over to be*
*clear and consistent.*

As mentioned above and by reviewer 1 (Roberti), we will check for consistency and refer to all deposits
as "supraglacial landslide deposits (>0.05 km$^2$)". We believe that many of the supraglacial landslides
are likely to be emplaced through a rock avalanche process, but, this is difficult to verify, and, for this
tool supraglacial landslide removes any process related issues.

**In line comments:**

*It may be worth mentioning in the main text that updates to RGI can be readily accommodated. I have*
*seen at least one announcement that RGI v7.0 has a release target by the end of 2020. This is indicated*
*in the supplement but not stated directly (but maybe it is obvious).*

We will update line 112 to read:

"Any updated version of the RGI will be incorporated when available. Additionally, the RGI can be
replaced by the user with shapefiles of the Greenland and Antarctic ice sheets, if analysis is required in
these regions, or higher resolution (user defined) glacier outlines, if the RGI is deemed insufficient."

*Line 35: Consider referring to point (i) as glaciological and climatological controls?*

Agreed. We shall change to: "(i) glaciological and climatological controls such as thrusting and meltout
of sub- and en-glacial sediment onto the surface (e.g. Kirkbride & Deline, 2013; Mackay et al., 2014;
Wirbel et al., 2018)"

*Line 55: "rapidly transported away from source areas" - in addition to rapid sequestration, which is I*
*think the point focused on in the sentence following the one where this is mentioned, is there a citation*
*about how runout extent of the event is different when deposited primarily on ice?*

We shall reword to "In supraglacial settings, landslides, where topography allows, travel much further
than their non-glacial counterparts due to the reduced friction of the ice surface (Sosio et al., 2012).
Rapid transportation away from source areas also occurs because of glacier flow. This removes the
simplest diagnostic evidence of a subaerial mass movement process – a linked bedrock source area and
debris deposit. Without the associated deposit bedrock source areas are easily mistakenly characterised
as glacial cirques (Turnbull and Davies, 2006)"

Sosio, R. et al. Modelling rock avalanche propagation onto glaciers, Quaternary Science Reviews, 47,
23-40, doi: 10.1016/j.quascirev.2012.05.010, 2012.

Turnbull, J.M. and Davies, T.R.H. A mass movement origin for cirques, Earth Surf. Proc. and
Landforms, 31(9), 1129-1148, doi: 10.1002/esp.1324, 2006.

*Line 60: Why use the term "censoring" here?*

We shall change to 'visibility'.

*Line 108: I would change this from "present day" since the RGI v6.0 was published in 2014 and likely*
*stops with digitized outlines before then*

We will change this to "1943 and 2014".

*Line 153: I'm not sure I understand the point that "GERALDINE is in effect standardised with this*
*global supraglacial cover map" - it would be help to expand on this point*

We are going to remove this now that an updated supraglacial debris cover map is available (Herreid
and Pellicciotti, 2020). We will reword to "We justify our 0.4 threshold based on Scherler et al. (2018)
who deemed it optimum for the creation of a global supraglacial debris cover map using Landsat data."

Herreid, S. and Pellicciotti, F. The state of rock debris covering Earths glaciers, Nature Geoscience,
doi: 10.1038/s41561-020-0615-0, 2020.

*Line 185: I had to read this sentence a few times. Maybe stating this in terms of candidate events*
*(instead of outputs) or being clear that identification step is the one that the user executes and that*
*GERALDINE only presents candidates? (See point above, as I'm advocating for a particular language*
*choice, just that it is a bit more clear and consistent)*

We shall reword to:

"Of the 48 validation RAs, the user was able to correctly identify 44 of these events from GERALDINE output maps, a true positive detection accuracy of 92 %. False negatives all pre-date 1991 (Figure 3), giving 100% successful user identification post-1991. These false negatives can be explained by a failure of Landsat satellites from imaging the RA deposit. This was due to reduced (and insufficient in this case) tier 1 Landsat image availability pre-Landsat 7 within the GEE data catalogue, inhibiting GERALDINE from highlighting the RA as new debris."

*Line 211: Is introducing the acronym SLC necessary? It is only used once (I think). In general there are a lot of acronyms (see comment below on Figure 1)*

We will remove this acronym.

*Line 255: Am I understanding this right that GERALDINE could not detect multiple landslide deposits in about the same spot but at different times? This may never (or only rarely) occur, but I wasn't sure if that was the point this sentence was trying to make. Or, something else about how the "user will have already determined the date of these earlier supraglacial landslides"*

We shall add a sentence at L228 addressing multiple failures:

"GERALDINE can also not detect landslide debris deposition onto an existing debris cover. Therefore, if a landslide consists of multiple failures, a GERALDINE output map would only detect one event, with the deposit extent being the combined total of all failures. It would be highly beneficial to combine GERALDINE with seismic detection to help delineate the amount of failures that occur."

Clarification of Line 255 will read:

"If GERALDINE is run annually for multiple years, the user will be able to determine the emplacement date for these earlier supraglacial landslide deposits."

*Section 3.3 - it seems like it would be worth mentioning that you can do this in the abstract. That would also help expand on the "monitoring" side of the tool's name up front*

We shall reword Line 17:

"GERALDINE outputs maps of new supraglacial debris additions within user-defined areas and time ranges, providing a user with a reference map, from which large debris inputs such as supraglacial landslide deposits can be rapidly identified and monitored. We validate the effectiveness of GERALDINE outputs using published rock-avalanche inventories. We then demonstrate its potential in Alaska by identifying two previously unknown, large (>2 km$^2$) supraglacial landslide deposits and track the evolution of an existing supraglacial landslide deposit."

*Line 285: What are the current methods that GERALDINE outperforms? Manual inspection of individual images? - Very very minor, but I also found that the original (and widely cited) paper by Ostrem has his last name typically spelled with a slashed O but in the original paper it is given with an umlaut. From my reading of this it may have been an older alphabet choice and that these are the same (https://en.wikipedia.org/wiki/%C3%96; https://en.wikipedia.org/wiki/Danish_and_Norwegian_alphabet). I just wanted to point out that the community overwhelmingly cites this paper with the author's name using a slashed O. And, subsequent work by Gunnar Ostrem uses the slashed O.*

We shall reword to:

"We showcase how GERALDINE does not suffer from the traditional disadvantages of current manual and seismic detection methods that can cause supraglacial landslides to go undetected, by identifying two new supraglacial landslides in 2018, in the Hayes Range of Alaska, one of which could not be detected using existing methods."

Thank you for pointing this out. We shall change the "O" in Ostrem to a slashed O.

*Figure 1: I would consider giving all these acronyms in the caption. Also, the second to last step isn't*
*quite clear - what is "both" here?*

All acronyms will be given in the caption. We shall change the second to last step to read "'Subtract $t^1$
debris map from $t^2$ debris map to highlight new debris" and add both $t^1$ and $t^2$ to the previous steps
where applicable.

*Figure 2: This figure made me look back to the text to make sure if I understood that the maximum*
*debris extent would merge the evolution of the event. I think that is true, regardless of the search*
*timeframe (and somewhat dependent on the Landsat image separation). This would mean that to track*
*the debris transport you would first find that an event occurred and then go back and look through all*
*images to characterize how it evolved - this is all a user step, right? I'm thinking of your Lituya*
*Mountain example: if you instead ran GERALDINE for the timeframe of 2012-2014 you would get one*
*maximum extent and then you would have to notice that the event occurred in 2012 and was still visible*
*in 2013 and 2014 frames. This is still great since it is relatively little work to analyze around a particular*
*event compared to finding the event in the first place. Right? I think some more context on how many*
*events may exist and how laborious it is search individual frames may help put this in context. And, you*
*could say a bit more about this workflow in Section 3.3, since what is said around line 260 isn't quite*
*clear how that connects to what is shown in Figure 2 (if at all).*

Yes, this would be a user step, and is why we suggest annual or sub-annual time frames. As mentioned
by reviewer 1 (Roberti reviewer response lines 77-89) on Line 233 we will add a section detailing the
time and computational savings GERALDINE makes vs manual inspection. We will reword Line 258
onwards to make it clear that this movement would be over the user-specified time period:

"A secondary use of GERALDINE is tracking existing supraglacial landslide deposits. These deposits
are transported down-glacier by ice flow, although often the initial emplacement geometry is
characteristically deformed and spread due to differential ablation and ice motion (Reznichenko et al.,
2011; Uhlmann et al., 2013). GERALDINE can give an indication of deposit behaviour and movement
by highlighting 'new' debris, at the lateral and down-glacier end of the deposit, as it moves between
image captures (Figure 2). Differencing the distance of this new debris from the previous year's deposit
extent can give an approximation of lateral spreading and glacier velocity over the user-specified time
period, the latter of which is often unknown at the temporal resolution of Landsat and complex to
calculate in high mountain regions (Sam et al., 2015)."

*Supplement: - Section 4.0 first paragraph should be "complementary" instead of "complimentary"*

We will change to "complementary".

**Sam Herreid (Reviewer 3)**

*The article "GERALDINE (Google earth Engine supRaglAciaL Debris Input dEtector) – A new tool for Identifying and Monitoring Supraglacial Landslide Inputs" By Smith at al. describes a tool that subtracts composite debris maps from two stacks of Landsat images, one from a period of interest and the other from the preceding year, to isolate new debris additions. A user can then interpret this output to locate supraglacial rock avalanches or landslides. I found the paper mostly easy to read and I think the research objective is timely and useful. I also appreciated the user guide provided in the supporting information. However, I think the authors stopped their tool development prematurely leaving some fundamental elements unaddressed. My main points of concern are briefly summarized here with more detailed comments inline below along with minor comments.*

We thank the reviewer for their time and comments. We have addressed each one of these below.

As an overarching response, the main purpose of GERALDINE is the rapid analysis of hundreds of Landsat images over large areas to aid in the detection of supraglacial landslide deposits, without the need for any computational/digital storage capacity or basic programming skills. This is a clear research gap. We have purposely designed the tool in this basic way using GEE, to keep outputs rapid, easy to use for those collating inventories, and viewable in a web browser, making it as accessible as possible. The validation shows the tool is fit for purpose, with few benefits, but many disadvantages of adding complexity at this stage.

We believe a number of the comments come back to the purpose of our work, to identify supraglacial landslides rapidly, in the cloud, and with non-remote sensing expert users, and, the reviewer expertise and recent publications on debris cover extents. We are not aiming (or wanting) to produce precise maps of all debris cover with minimum noise, we are aiming to detect slope process inputs that are usually time consuming to identify, or, not identified at all.

*Looking at the two map figures of the article, it is clear that, even within the GEE stack methodology, which is in principal sound, debris cover is not confidently mapped. There is unphysical debris in the accumulation zone and many instances of "new debris additions" that are not new debris additions. These areas accumulate into tool output false positives that are neglected by the authors who rather only report true positive success, leading to statements like L283-284: "GERALDINE outputs [had a] 100% successful identification". By neglecting to calculate a metric like precision or the false positive rate, the study is lacking a meaningful assessment of performance. I think it is reasonable to state, as the authors do, that some of these debris map errors stem from errors in the RGI, but these then need to be either mitigated or quantified in the error assessment of your tool.*

We have not designed GERALDINE to map all supraglacial debris cover in the most accurate and confident way, that would require a large amount of performance enhancements and accuracy assessments, as the reviewer rightly points out, and is beyond the aims of this research. Using the image stacking method, GERALDINE finds the maximum debris extent. This approach would be unsuitable for accurate debris cover mapping, as any temporally inconsistent/misclassified debris pixel is amplified into the final debris mosaic, evidently creating some debris false positives. To map global debris cover in an accurate way (which is not the aim of our work), an average approach would need to be used, which has been done elsewhere (e.g. Scherler et al. 2018). Using the maximum debris extent does, however, allow supraglacial landslides to be detected effectively, and is particularly useful for those with a short surficial residence time e.g. landslides in accumulation zones. It is therefore wrong to think of GERALDINE as a tool to accurately map all supraglacial debris cover. Instead it should be used as a tool to highlight new possible supraglacial landslide deposits, which is not often done. We agree that it would be optimal to do a validation in which we could quantify all true/false positives/negatives, with an error matrix and associated statistics. However, due to the way the tool gets a maximum debris extent using the image stacking method (if just one pixel in the image stack is debris, that pixel in the final mosaic will be debris), there is no dataset we can use to perform such a validation. All existing datasets rely on an average or singular image to calculate debris coverage, which is completely unsuitable for validating GERALDINE outputs against. We have confidence in outputs though because the underlying image classification methods (cloud removal and band ratio algorithms) work, as they have been used and peer-reviewed elsewhere. We have therefore undertaken a validation in this way to provide some measure of supraglacial landslide detection accuracy and believe it is suitable for these purposes. With regards to allowing a user to detect supraglacial landslides, our 100% successful user identification of validation RA deposits post-1994 is valid, as an expert in slope processes was able to successfully identify 100 % of them from GERALDINE output maps. To begin a discussion on the tool being used beyond its purpose, and it failing to do that, is not of benefit here.

*From my view, the main incentive for a tool that considers every image acquired in a stack, is to detect rock avalanches that are deposited onto a glacier's accumulation zone and automatically assign a best constrained deposition date. The automated detection of rock avalanches deposited onto bare glacier ice in ablation zones is also useful, but there is less chance of missing one since there will be a surface expression in every snow/cloud free image after deposition until it is too heavily reworked or evacuated from the glacier. Further, in ablation zones there is the case, that will likely only grow in frequency, where a rock avalanche is deposited onto existing debris cover, or earlier deposited rock avalanche debris, which is an entirely undetectable event using this method. By summing debris cover over one year or longer, the method presented here will likely catch a deposit onto the accumulation zone, but by not finding the difference between each sequential image the approach loses any ability to assign a deposition date. I understand the incentive to aggregate debris, but from the comment above, I think the quality of the resulting debris maps are still low relative to other automated debris maps in the literature.*

Although it is relatively easy to spot landslide deposits in glacier ablation zones by viewing individual images, GERALDINE eliminates the need to manually analyse the entirety of 22+ Landsat images, making it 22x less time consuming for a user (any one Landsat sensors capture 22 images per year, at a frequency of every 16 days. Except for 1993-1999, there are always two Landsat sensors imaging the earth surface, making it likely that there are 44+ images every year). GERALDINE outputs are also characterised by high contrast between new and old debris, making it much easier to identify supraglacial landslide deposits and narrow the window of event occurrence. As reviewer 3 points out, the main benefit of GERALDINE is the ability to spot supraglacial landslides deposited in accumulation zones, which have thus far not been quantified well. With regards to debris deposition on existing debris cover, this is a limitation of the tool. However, we argue that identifying debris onto clean ice (i.e. expanding debris cover) is of greater importance than debris onto existing debris, which is likely to have a much lower impact on reducing glacier melt, and only affects the glacier through  mass input (i.e. accumulation). It would of course impact frequency-magnitude estimates over these portions of a glacier. We agree this issue may grow if debris cover expands over most glacial areas (as the reviewer has just published on). The ability to assign a deposition date is an element we have tried to incorporate to GERALDINE but by using a mosaic derived from an image stack, the metadata of pixel date/time used is lost, so it is not possible using our current approach. A user would typically always want to view the 'raw' time-stamped imagery that created the image stack after a positive supraglacial landslide deposit ID, to investigate its characteristics. Determining an event date and time adds little workload to this procedure. As mentioned previously, we know our debris maps are of lower quality than others in the literature, but, again, that is not the aim of our tool. Rather than trying to accurately map the entire debris cover, we aim to quickly highlight where potential supraglacial landslide events have occurred. Using GERALDINE this task can be done easily by a user (especially those without expert remote sensing abilities, but, with slope process expertise) without considerable resources, as we eliminate the need for any specialist computing/storage/programming requirements.

*I think a rock avalanche deposit onto bare glacier ice is a strong signal that can be detected automatically. For example, the area of a rock avalanche feature will almost always be much larger than any other location of debris additions from other sources (if mapped accurately and dt is short, e.g. 1 year). The authors leave this step to the user which I think significantly reduces the applications of this tool. I can accept that this version does not need to perfectly resolve all of elements to mapping rock avalanches onto clean glacier ice, but I think providing an automated selection of rock avalanches from new debris additions is only a minor addition that will increase both the tool application as well as ability to quantify true positives, false positives and false negatives. I also think that looking at the differences between every image after a rock avalanche is detected to constrain the date of deposition is a reasonable and achievable result at this stage of tool development.*

In an ideal world we would provide some form of automatic detection of supraglacial landslide deposits, but their size and shape vary so considerably on glaciers, particularly in steep, meandering glacier terrain, with frequent cloud cover, that this was decided as unfeasible for the initial tool creation. It would result in many missed deposits – especially for those where snow and ice entrainment make the morphology complex. Any threshold on size and shape would almost certainly lead to some supraglacial landslide deposits being overlooked. We do envisage this to be a future part of tool development, but this initial version is already vastly superior in speed and processing requirements than other methods, so we want it to be available to the community as quickly as possible in its current form. We also feel that only basing the tool on existing validation inventories trains bias into any auto-detection, we wish GERALDINE to be run with the semi-automatic approach to derive key features that may allow automation of detection. With regards to looking at the differences between every image, this is possible and earlier versions of GERALDINE did utilise this method. However, cloud dominated images would have large areas masked out, therefore differencing them with any other image provides no useful comparison, due to a lack of data from which to difference with. Using an annual stack of images gives a solid baseline from which differencing can be undertaken. In addition, we would still advise manual validation using original Landsat imagery, so this image differencing was deemed unnecessary, adding additional processing time and user interaction with the tool. It is also likely that a user will want to determine supraglacial landslide source areas, which would also require viewing the original Landsat imagery.

*If this method is to be a starting point for a globally applicable tool (L22), I am concerned that the authors cite limitations of GEE that cause the region of interest to be limited to <5000 km2. Do the authors anticipate that this method could be written in a more computationally efficient way such that this limit will be dramatically increased? Highly useful functionality of a tool like this one will be when all of Earth's glaciers can be assessed in near real time, but if there are intrinsic limitations within GEE is this a feasible future for this tool?*

GERALDINE has been through multiple iterations to ensure the code is written in the most computationally efficient way possible with the current method (to our knowledge). It can handle much bigger areas than 5000 km$^2$ but processing is significantly slower, as is panning/zooming around the map in the browser, due to the way GEE computes these layers on the fly. If calculating large areas, the best approach is to export them and view them in a GIS. We will explain this in the methods section of the manuscript by rewording from Line 116 onwards:

"GERALDINE gathers all Landsat images from the user-specified date range and the year preceding this user-specified date range, within the user-specified region of interest (ROI), creating two image collections within GEE. Users should note that smaller ROIs and annual/sub-annual date ranges increase processing speed, with processing slowing considerably with >800 Landsat images (~160-1500 GB of data). The software clips all images to the ROI, applies a cloud mask, then a water mask, before finally delineating supraglacial debris cover from snow and ice. GERALDINE acquires the maximum debris extent from both image collections, creating two maximum debris mosaics, then subtracts these mosaics and clips them to the RGI v6.0 (or user defined area if not using RGI) to output
a map. This map highlights debris within the user-specified time period that was not present in the
preceding year, which we term 'new debris additions'. This map is viewable within a web browser as a
layer in the map window. However, as it is calculated 'on-the-fly' (Gorelick et al., 2017), large areas
can be slow to navigate. All files can be exported in GeoJSON (Georeferenced JavaScript Object
Notation) format for further analysis, including to verify if detections are discrete landslide inputs. This
is recommended for large ROIs. An overview of the workflow is presented in Figure 1 and the detail
for each step described in Sections 2.1.1–2.1.4."

*Finally, there is a factor present in the quantity "new debris additions" that is not quantified or*
*discussed. Unstable glacier flow will produce debris structures that deviate from flow lines parallel to*
*a glacier's valley wall (e.g. the surge loops on Susitna Glacier in your Figure 4) and a difference map*
*of debris cover over some dt will show a false gain and false loss of debris cover that is really just*
*debris structure translation. Where glacier flow instabilities are present, a simple difference of debris*
*cover maps cannot be strictly new debris additions. Herreid and Truffer, 2016 provides a discussion*
*on this topic. Herreid, Sam, and Martin Truffer. "Automated detection of unstable glacier flow and a*
*spectrum of speedup behavior in the Alaska Range." Journal of Geophysical Research: Earth Surface*
*121.1 (2016): 64-81.*

We thank the reviewer for raising this interesting observation. This is a valid point if the use of
GERALDINE was for accurate debris cover maps, but, as mentioned previously, we are only interested
in supraglacial landslide deposits. If a user familiar with glacial landslides/glacial flow was to view
these areas classified as new debris, they would be able to determine straight away that these were not
supraglacial landslide deposits, due to their size and shape. However, we will add a section about this
on line 277 as these features can be seen in figure 5:

"We note other areas are flagged as 'new debris' in 2013 and 2014. These are typically where glacier
downwasting has occurred exposing more of the valley walls, or where there has been temporal
evolution of the debris cover i.e. glacier flowline instabilities. These flow instabilities can cause double-
counting of debris when larger time windows are specified (described further in Herreid and Truffer,
2015). Both processes subsequently cause false classification as 'new debris', however, neither display
supraglacial landslide characteristics, so it is highly unlikely a user would mistake them for one."

Herreid, S. and Truffer, M. Automated detection of unstable glacier flow and a spectrum of speedup
behaviour in the Alaska Range, Journal of Geophysical Research: Earth Surface, 121(1), 64-81, doi:
10.1002/2015JF003502, 2016.

**In line comments:**

*L1: Perhaps stylistic but I think "A new tool for identifying and monitoring supraglacial landslide*
*inputs" is a better title, without the less straightforward and somewhat redundant acronym.*

We thank the reviewer for proposing an alternative title, but remain with our original wording because
the tool name is key to identifying its purpose and is more memorable for a user – similar for example
to the well-known 'Google Earth Engine Digitisation Tool (GEEDiT)'. Furthermore, it includes the
name of the platform – Google Earth Engine – where the tool is executed.

*L9: Why not use "rock avalanche" throughout? I believe rock avalanche is more precise and consistent*
*with the literature for what you are looking at. If the authors prefer the more general term landslide,*
*then early in the introduction make clear what is and is not a landslide vs rock avalanche for this study*
*and keep the language consistent. It's strange to read landslide in the title and have rock avalanche be*
*the first sentence of the abstract.*

We agree that terminology is not consistent throughout. As per our response to reviewers 1 and 2, we
will be much clearer about this in the text and reword sections of the manuscript referring to all large
debris inputs detected (>0.05 km$^2$) as "supraglacial landslide deposits". This is because we do not know
the processes which resulted in slope failure (although for many, and within the validation data set they
are almost certainly of RA origin), and, as reviewer 3 rightly points out, changing between different
terms is confusing. We propose that one umbrella term, e.g. supraglacial landslide deposit, will address
this. However, we do validate GERALDINE against RA deposits, as these examples have been
investigated, confirming their failure/deposition process. We will reword the abstract introduction to:

"Supraglacial landslides are high-magnitude, long runout events, believed to be increasing in frequency
as a paraglacial response to ice-retreat/thinning, and arguably, due to warming temperatures/degrading
permafrost above current glaciers."

*L9-12: There is a missing step here, rock avalanches can happen far from glacier ice. Detection of RAs*
*for the study of RAs alone, or to answer frequency questions with respect to climate or ice factors,*
*should consider all RAs independent of their runout happening to be on a glacier. This is either a very*
*big sampling bias or you should pose a glacier specific problem.*

With our terminology updated to "supraglacial landslides", we believe this section is now clear at
proposing a glacier specific problem. The purpose of this paper is to exactly fill the sampling/spatial
bias you refer to. Off glaciers subaerial landslide deposits have far longer residence time in landscapes
and there is less likely to be under-detection, although this can vary depending on how rapid geomorphic
processes are.

*L14: It reads like you are focusing on filling this small to medium gap but on L43 you say you focus on*
*the inputs of high magnitude, > 10ˆ6 mˆ3, RAs. Please clarify/fix and keep consistent throughout. L215*
*considers a 0.062 km2 event.*

As per above, all terminology is to be changed to "supraglacial landslides", with the tool an aid to detect
"supraglacial landslide deposits (>0.05 km$^2$)". As per our response to reviewers 1 and 2 we will reword
L43 to:

"Here we focus on supraglacial landslide deposits (>0.05 km$^2$), commonly associated with RAs, defined
as landslides: (a) of high magnitude (> $10^6$ m$^3$); (b) perceived low frequency; (c) long runout; and (d)
where there is disparity between high present-day rates of slope processes above ice (Allen et al., 2011;
Coe et al., 2018) and expected rates based on theories of lagged paraglacial slope responses (Ballantyne,
2002; Ballantyne et al., 2014a)."

*L22: From the abstract alone you don't mention measuring area or volume or event timing, so I don't*
*quite see the jump to a global product. Further, on L118 you advise ROIs <5000 km2. Do you anticipate*
*a less computationally costly version of your method or are there HPC options in GEE? Finally, it is a*
*little strange to have a first step towards a revision, a revision implies several steps have already been*
*taken.*

The sole purpose of GERALDINE is to identify supraglacial landslide deposits – mentioning area,
volume or event timing in the abstract implies greater tool capabilities than it has, as these are manual
steps. We are clear that the tool only produces maximum debris cover maps. As mentioned in above
comments, it is possible to run the tool with areas >5000 km$^2$. We shall substitute 'revised' to
'complete'. Volume requires area-volume scaling relationships that are uncertain, and timing within
Landsat repeats is best done with a focussed search through seismic data (as we are doing in
collaboration).

*L26: With the known errors in the RGI, it's better to avoid presenting the number of glaciers to the*
*accuracy of a single glacier. Consider ">200,000".*

Agreed. We shall amend this.

*L27: Consider a revised global estimate of debris cover from Herreid and Pellicciotti, accepted by*
*Nature Geoscience, which should be available by August 2020 at this DOI: 10.1038/s41561-020-0615-*
*0*

We shall change this to read "Recent estimates suggest supraglacial debris only covers 7.3% of the area
of this glacier (Herreid and Pellicciotti, 2020), up from 4.4% estimated by Scherler et al. (2018). For
many glaciers…"

*L34: Either add "e.g." to the citations or also add a citation to Kirkbride and Deline, 2013 whose Table*
*1 gives a more complete list of citations for expanding debris cover. Kirkbride, Martin P., and Philip*
*Deline. "The formation of supraglacial debris covers by primary dispersal from transverse englacial*
*debris bands." Earth Surface Processes and Landforms 38.15 (2013): 1779-1792.*

We shall add a reference to Kirkbride and Deline (2013).

*L35: What is the difference between sub- and en- glacial sediments in this context? I don't think sub-*
*glacial sediments can melt out.*

We cite Mackay et al. (2014), who provide evidence from Antarctica that subglacial sediment can melt
out. In their case much of this debris were rockfalls that entered in the accumulation area and reached
the basal zone. We'd also direct the reviewer to the literature on Blue Ice Moraine where subglacial
debris stores are brought to the surface by compressive flow and melt out as distinct debris bands.

*L35: Anderson, 2000 addresses general dispersion of medial moraines which you don't explicitly*
*mention here. Does "debris store" mean extraglacial debris? This is not clear. Anderson, Robert S. "A*
*model of ablation-dominated medial moraines and the generation of debris-mantled glacier snouts."*
*Journal of Glaciology 46.154 (2000): 459-469. L36: It might be worth distinguishing here high volume*
*low frequency mass movements from low volume high frequency.*

We shall add "(ii) dispersion of medial moraines (Anderson, 2000)" and subsequently shift ii to iii and
iii to iv. We shall reword to "(iv) remobilisation of ice proximal, extraglacial debris stores, particularly
lateral moraines (Van Woerkom et al., 2019)."

On line 43 we will discuss magnitude frequencies:

"Magnitude-frequency relationships suggest these low frequency, high magnitude events have a
disproportionate effect on sediment delivery (Malamud et al., 2004; Korup and Clague, 2009). One of
these large events mobilises enough debris to dominate overall volumetric production and delivery
rates, exceeding that of the much higher frequency but lower magnitude events."

Korup, O. and Clague, J.J. Natural hazards, extreme events and mountain topography, 28(11-12), 977-
990, doi:10.1016/j.quascirev.2009.02.021, 2009.

Malamud, B.D. et al. Landslide inventories and their statistical properties, 29(6), 687-711, doi:
10.1002/esp.1064, 2004.

*L43: How are you able to focus on landslide of a particular volume? Throughout you do not calculate*
*or consider volumes. And do you mean high volume? Magnitude of what?*

This is a fair point, and as mentioned above in previous comments (Line 630-645 of reviewer responses)
we shall remove the $10^6$ m$^3$ volume from this section and define what we are interested in as
"supraglacial landslide deposits (>0.05 km$^2$)". Volumes require scaling laws. Part of this goes back to
the RA process identification, which by definition involves over $10^6$ m$^3$ volumes.

*L44: "where there is disparity between current high rates of activity above ice" this is unclear.*

Recent research cited evidences high rates of RA activity in glacial environments, but it is expected that this response is typically delayed until deglaciation (see Ballantyne references). We believe this is clear if the full sentence is quoted.

*L46: lag ice-free conditions in terms of what?*

Theory of delayed slope response to deglaciation. We shall change to "lagged paraglacial slope responses since deglaciation (Ballantyne, 2002; Ballantyne et al., 2014)".

*L47: What does "relatively low in the landscape" mean?*

We shall amend this to "relatively low elevations in the landscape".

*L58: I'm not sure if there are remote sensing methods yet to see englacial debris. Maybe you mean geophysical methods, e.g. GPR.*

Operation Icebridge data can image englacial debris, and these data are sensed remotely – this is a language/discipline point- many publications use for example, 'GPR Remote Sensing in Archaeology' (Springer) where a geophysical technique is used to remotely sense a target. We shall reword to "non-ice-penetrating remote sensing and ground-based techniques".

*L59: "[add: potentially] considerable modification"*

We shall apply this change.

*L60: "Deposited"? "Emplaced" is odd. L72: Landslides vs RA confusion here.*

We shall change to 'deposited'. See earlier comments (Line 630-645 of reviewer responses) r.e. Landslide/RA confusion and continuity.

*L87: Open access or open source?*

After consideration we think it is wrong to describe GERALDINE as either open access or open source. Google Earth Engine requires a sign-up, so it is not 100 % open access and you cannot access the underlying code of certain functions, so it is not truly open source. We shall change the text to reflect this by rewording mentions of open access to "free-to-use" and remove any mention of open source.

*L90: Define what you mean by "wide" in parentheses*

We shall substitute 'wide' for 'large'. It is however difficult to quantify because it depends on the extent of glaciers in the region, the amount of Landsat images to be processed and whether a user wants to view it in a web browser or export it and view it in a GIS. For example, the study area could be $10^6$ km$^2$ and have $10^3$ km$^2$ of glacier coverage and run fine, but a study area of $10^4$ km$^2$ with $10^4$ km$^2$ of glacier ice could cause processing issues. GERALDINE can struggle to display the results of larger areas with >800 images within browsers, as they are calculated on the fly, as explained on Line 584-598 of our response to reviewer 3 comments. However, if this layer was exported and viewed in a GIS, there would be no issue. As mentioned on line 551-577 we will add this information to the method section.

*L109: RGI errors are further quantified in Herreid and Pellicciotti, accepted by Nature Geoscience, available around August 2020 at DOI: 10.1038/s41561-020-0615-0*

We shall cite this study, in addition to Scherler et al. (2018).

*L116: add: "[and all images in the] year preceding. . ."*

We shall adopt this change.

*L118: What do you mean by "specify annual date ranges"? Are you saying the tool can only work for one time window between two specified years? This seems like a pretty critical limitation to the*

*functionality to the tool. Are you sure GEE is the correct platform if its memory capacity is such a bottleneck? Maybe JuypterLab is a better cloud-based platform? Or your code could select a single optimal image of a one year stack and then make your calculations on single images? Also if you clip the RGI first, then all of your calculations will be less computationally costly.*

Annual date ranges, or less, are the optimum time ranges to use. The tool can work for as many years as a user wants but the outputs are affected, due to the way GERALDINE retrieves maximum debris extent. As mentioned previously, any artefact is amplified into the final mosaic, so if run over 10 years, there would be 10 years of artefacts in the final debris mosaic. We also see no reason to run over multiple years, as the metadata cannot give a deposit date/time. Running over annual ranges not only improves the visibility of supraglacial landslide deposits (due to less artefacts), it also narrows down the window of occurrence, making it easier for a user to determine deposition dates with GEEDiT.

We are confident GEE is the correct platform for the tool because it is a familiar environment for a user with no experience of programming, it is free for researchers, has a large data catalogue and has suitable computational capabilities, allowing for further development. With regards to optimum images, these would ruin the ability of the tool to detect deposits which occur in accumulation zones and are consequently only visible in one image. See lines 563-571 of our responses for why a single image method was not applied.

Although we welcome suggestions regarding tool efficiency, clipping to the RGI first is in fact much more computationally costly. Earlier versions of GERALDINE processed images by clipping to the RGI first, as we came to the same conclusion, but clipping is a memory intensive task in GEE, and the RGI has thousands/millions of vertices. This made clipping every image to the RGI pre-analysis, 75% more memory intensive and subsequently 60% more time intensive.

*L122: This section is not very clear, but if I understand correctly, the tool will collect two stacks, one from the year before a defined date range and one for the full defined date range, and then perform a single subtraction to find a single map of new debris. There is an issue of accumulating "new debris additions" if the stack of images aggregate debris from, say, 10 years, there will be much more new debris additions that are not sourced from RAs. You also lose the ability to automatically detect a deposition date which is, in my view, the main incentive to use GEE and consider stacks of images rather than single optimal images. I think maybe you should change the wording of a "user-specified date range", and rather say "a user specified year where the tool will give you a map you can look for RAs deposited since the preceding year." But 1. I don't understand why finding the RAs can't also be automated, this should be a very clear signal if deposited on clean ice (you will entirely miss RAs that are deposited onto existing debris cover); and 2. As a user I can think of two uses for a tool like this: (a) getting the location of all RAs that have been deposited onto a glacier and are still present at the surface and a deposition date if deposited since Landsat 4; and (b) near-real-time detection. I think your tool could be successful for the latter, although to be practical it should be able to analyze all of Earth's glaciers at once or at least all glaciers in, say, Alaska (Bearing Glacier in SE Alaska alone is larger than the recommended <5000 km2 ROI), but I think there is still a lot of improvement needed for the former. The difference map needs to be computed annually to keep other debris addition signals small and also facilitate a deposition date.*

Please refer back to our responses on lines 543-598 about single images, large ROIs and automatic detection of RAs. We disagree with the "user-specified date range" word changing because the tool has two main uses: (1) finding supraglacial landslides on an annual/sub-annual basis; and (2) finding very recent RAs. For example, if a RA derived seismic signal has been detected, this seismic signal can only locate to within a 100 km$^2$ radius. A user can then utilise GERALDINE, specify a short date range, and incorporate real-time Landsat imagery within this 100 km$^2$ radius, to identify its location. As mentioned in our response on lines 572-598, we will amend and expand on the ROI size requirements, to make it clear that larger areas are fine, and explain the caveats that come with increasing a ROI.

*131: I can appreciate that the method used to assess cloud mask performance considers clouds in an entire stack, thus incorporating a variety of cloud types in a simple run of your code. However, I would like to see more direct evidence that clouds themselves are accurately mapped. From my experience cloud mapping algorithms are unreliable in glacierized areas. Could you show a side by side image of a raw satellite image and an overlay of the output of the cloud mask with scores 20%, perhaps one where it worked well and a second where it was at its worst. I'm concerned that you're only mapping 60% of RA area. How were the studies that make up your validation dataset able to map 100% of the RA area and you cannot? Surely with the stack methodology the aggregate over many images should, together, capture 100% of RA area unless it's a particularly snowy or cloudy year. Does this suggest you have a 40% error rate in detecting RAs?*

We have found the GEE in-built cloud mask to be surprisingly good in glacierised regions, but a side-by-side comparison of a good and bad example is a great idea, which we will include in the supplementary information. As mentioned in the manuscript (line 170), we use area as a proxy for how easy it is for a user to identify these deposits. It is unlikely that a 100% deposit area detection could be achieved because of the way supraglacial landslide deposits are often partially advected into the ice and unpredictably entrain snow and ice during transport. If Landsat imagery does not image a deposit within hours/a few days of occurrence, it is highly likely that a 100% deposit area detection is unachievable with the available Landsat imagery. Some of the validation dataset utilised RAs of this nature, hence the average 60% area detection. Many manual detections are not 100% of a supraglacial landslide deposit, they are an interpretative map which is more difficult with increasing time from deposition, and, we are attempting to validate against these often imperfect (not a criticism) data.

*L132: What about cast shadows from topography? Herreid and Pellicciotti, accepted by Nature Geoscience (available August 2020 at DOI: 10.1038/s41561-020-0615-0) found it necessary to remove area in shadow in order to accurately map debris cover. The band ratio method is able to negotiate some shading, but when a surface becomes too dark there is still the possibility for false positive debris classification (e.g. Herreid and Pellicciotti, 2020 removed 760 km2 of shaded glacier area in Alaska and Western Canada).*

As mentioned at the start of our responses to reviewer 3, the method of GERALDINE is unsuitable for accurately mapping all supraglacial debris cover. As we explain for cloud shadow in the manuscript, masking shadow has minimal effect on the user's ability to identify supraglacial landslide deposits, whilst greatly increasing processing complexity and time (L132 of the manuscript). Any shadow is highly unlikely to be lobate and elongated; the typical characteristics a user would look for in a supraglacial landslide deposit. After running GERALDINE for a 90,000 km$^2$ area of Alaska on an annual basis from 1984-2019 (results not presented here), we have not had one instance (to date) of supraglacial landslide misidentification because of topographic or cloud shadow.

*L134: I don't really see a justification for the step of mapping supraglacial lakes or ponds. These features generally develop in heavily debris-covered portions of glaciers where your tool will fail to detect a RA by not having the prior bare ice context. Further, if these features are 22 pixels on average, as you cite in the SI, then the above discussed 40% omission error dwarfs the stream/pond signal. If you elect to keep this component please provide an example in the SI that shows how mapping streams and ponds leads to a higher rate of RA detection.*

We thank the reviewer for raising this interesting point. Based on these comments and on reflection, we agree that mapping of supraglacial lakes/ponds is unnecessary. We originally implemented it to reduce misclassification of new debris, but these are so small it is not necessary. A landslide deposit would/could cover any lake/pond during deposition and lakes display no supraglacial landslide deposit characteristics, so misclassification/misidentification is not an issue. We have determined it has no effect on the detection results and will therefore remove this from the code and manuscript.

*L150: One of your inequality signs should include "or equal to"*

We will amend this to "and snow/ice (≥0.4)".

*L158: There is a missing discussion on double counting translated debris features that deviate from a*
*flowline parallel the glacier valley walls. Also summed non-RA debris additions if the user defined time*
*period is not sufficiently short. Herreid and Truffer, 2016 established a very similar methodology to the*
*one presented here in order to detect glacier flow instabilities. In this study RA are identified but*
*considered an error in the context of the flow instability research question. For your work, RAs are*
*signal and the features identified by Herreid and Truffer, 2016 are errors. These should be discussed.*
*Herreid, Sam, and Martin Truffer. "Automated detection of unstable glacier flow and a spectrum of*
*speedup behavior in the Alaska Range." Journal of Geophysical Research: Earth Surface 121.1 (2016):*
*64-81.*

On line 277 we will add:

"We note other areas are flagged as 'new debris' in 2013 and 2014. These are typically where
downwasting has occurred exposing more of the valley walls, or where there has been temporal
evolution of the debris cover e.g. glacier flowline instabilities. These flow instabilities can cause double-
counting of debris when larger time windows are specified (described further in Herreid and Truffer,
2015). Both processes subsequently cause false classification as 'new debris'; however, neither display
supraglacial landslide characteristics, so it is highly unlikely that a user would mistake them for one."

Herreid, S. and Truffer, M. Automated detection of unstable glacier flow and a spectrum of speedup
behaviour in the Alaska Range, Journal of Geophysical Research: Earth Surface, 121(1), 64-81, doi:
10.1002/2015JF003502, 2016.

*L162: What do you mean by "Debris biased"?*

We think it is clear what is meant by this, as the previous two sentences explain how debris always
takes precedence over snow/ice in the final mosaics. We will amend so the user is directed once again
to Figure 2: "GERALDINE is therefore debris biased due to this processing step (Fig. 2)".

*L168: Do you mean an omission/commission validation? If not, please provide an additional sentence*
*on why a bipartite approach was used.*

The validation was undertaken in two stages, so we will reword to 'A two-stage validation was
undertaken…"

*L172: RA already defined.*

We shall change to read "Validation was performed against the already defined supraglacially deposited
RA databases of…"

*L175: 48 suitable events were found out of how many that you considered? It is helpful for the reader*
*to know if these are rare occurrences or the majority. I assume these inventories only consider*
*supraglacial RAs?*

We shall update this to reflect the total number of RAs in these databases. We will reword this sentence
to:

"Forty-eight events out of a total of 325 met these criteria, their locations distributed across the
European Alps, Alaska, New Zealand, Canada, Russia and Iceland."

*L175: please add a map figure showing all of the regions you applied your tool*

We agree that this would be a useful addition and shall add an additional figure in the supplementary
information with the locations of all validation RAs depicted on a world map.

*L189: I think if your code mapped RAs from the best available image for each event, rather than a*
*composite, you could be very close to 100%.*

Please see earlier comments (lines 543-598 and 802-830 of our response) as to why this method was
not used. We want to exploit all imagery, increasing our chances of detecting supraglacial landslide
deposits with short surficial residence times. These deposits may only be visible in one Landsat image
and are commonly missed by manual imagery analysis. This is due to time constraints only allowing
analysis of one or two optimum images in a year, particularly over large areas. Over 20+ images can be
discarded annually because of this, all of which may contain new supraglacial landslide deposits. This
is particularly true in images with high-percentage cloud cover that are commonly discarded from
manual analysis, but in the rare cloud-free areas of the image, may contain new, unknown deposits,
with short surficial visibility to optical remote sensing. GERALDINE can exploit all data from these
images that are typically discarded, making it a valuable time-saving tool for a user identifying
supraglacial landslide deposits.

*L189: A relevant factor that you do not mention is a RA that crosses existing debris cover. This is likely*
*the predominant factor of why you will not be able to map RAs to 100%.*

We shall amend this sentence to read "However, a true 100 % detection rate for supraglacial landslide
events on glaciers is unlikely, due to some deposits running out over existing debris cover, and some
having high snow/ice content or entraining large amounts of snow/ice during events, which can be
common for supraglacial landslides deposited onto glaciers."

And as mentioned in our response on lines 381-392, we shall add a sentence at L228 addressing multiple
failures:

"GERALDINE can also not detect landslide debris deposition onto an existing debris cover. Therefore,
if a supraglacial landslide consists of multiple failures, a GERALDINE output map would only detect
one event, with the deposit extent being the combined total of all failures. It would be highly beneficial
to combine GERALDINE with seismic detection to help delineate the amount of failures that occur."

*L196: The accuracy of the satellite image remains the same, the overall significance of a single pixel*
*of a small glacier increases.*

We shall amend this sentence to read "This is particularly applicable to small (<0.5 km$^2$) glaciers, where
the overall significance of a single pixel increases."

*L197: Looking at the noise in bare ice regions of Figure 4 I struggle to see what you mean by "true*
*negative detection rate is also extremely high"*

We refer to earlier responses explaining that GERALDINE is not a tool to accurately map all debris
cover. Despite this, noise is apparent in regions where temporary surficial debris cover is likely (so
classification as debris would be correct) i.e. at the bare ice-debris interface, and/or there are
discrepancies with the RGI i.e. where surface lowering has occurred, exposing more nunatak/valley
highlighting "new debris" around them.

*L198: I don't agree with this justification for user verification. If you subtract two optimal satellite*
*images before and after a RA deposition onto a non-debris-covered portion of a glacier, the signal is*
*exceptionally prominent, and I see no reason why an algorithm cannot easily identify this automatically.*
*I think somewhere in your GEE stack processing, the debris mapping and the cloud removal, a very*
*clear signal becomes muddy. I think some small changes to your workflow can provide a much clearer,*
*and likely more computationally efficient output.*

Please see earlier comments (lines 899-911 of our response) as to why this method was not used. Two
optimal satellite images would certainly provide a prominent signal but then you lose a large amount of
temporal data, which is crucial for detecting supraglacial landslides in accumulation zones that may
only appear in one Landsat image. We have developed the tool to use every available cloud-free pixel,
to extract the maximum amount of potential debris from an image stack. The image stacking method is
what makes the tool unique and allows it to extract the maximum amount of debris information. We
disagree that this creates a 'muddy' signal because supraglacial landslide deposits are always easily
identifiable to a user. We therefore stand by this statement of justification.

*L199: The problem with saying "to a user familiar with glacial and landslide processes, the [tool output*
*is] clear" is that a user familiar with glacial and landslide processes will be able to spot large*
*landslides onto bare ice from a raw image. The spatial domain of the tool is low<5000 km2 and the*
*tool cannot iterate over many years to pinpoint a deposition date. I think there is a lot of potential in a*
*tool like this but in its current state I have a hard time seeing a scientific application.*

This is true. However, as explained previously, for any one location there are 22+ raw Landsat images
in a year. Some of these may be neglected by a user because of high cloud cover, but these neglected
images may contain new supraglacial landslide deposits that have been advected into the ice by the time
the next image has been captured. The purpose of GERALDINE is to allow a user to aggregate and
extract all supraglacial debris information from every Landsat image, within their timeframe of interest.
As above (line 572-598 and 746-754 of our responses), we shall add a paragraph to better define the
spatial domain of the tool and how it can be run over much larger areas, depending on certain variables.
From our experience of using the tool in Alaska, we know that it allows a user to drastically improve
upon existing RA inventories, with current underestimation from initial analysis suggesting 50% of
supraglacial landslides are not found by manual analysis of raw images (manuscript in prep.). The major
point also remains, the time and computing capacity (both processing and storage) saved in looking
through GERALDINE outputs versus raw image investigation is considerable, and, is a large scientific
justification.

*L202: Please add a section to methods describing how your derived areal extent. Presumably there was*
*a manual step involved in this.*

Please see L177-182 of the manuscript. We shall reword to:

"GERALDINE was run for the year of the event using Landsat tier 1 imagery; the new debris vector
output file was exported into a GIS and after an initial qualitative step to see if the user would flag the
RA from the GERALDINE output, the area of the deposit it detected was calculated within the GIS."

*L215: How much user interpretation was involved with isolating the 71% true-positive RA area? False*
*positive and false negative areas must also be considered to make a statement about detection*
*confidence.*

No user interpretation was involved with isolating the RA area. We utilised the select by location tool
in QGIS, to select any pixels/pixel clusters within/intersecting the digitised RA polygon, and clipped
these pixels to the RA polygon, before calculating their area. We shall amend L180 to read "We utilised
the select by location tool in QGIS, to select any pixels/pixel clusters within/intersecting an outline of
the RA manually-digitised from a Landsat image using the Google Earth Engine Digitisation Tool
(GEEDiT) (Lea, 2018). We clipped selected pixels to the manually digitised RA outline and calculated
the area of these selected pixels."

We refer back to line 471-504 of our reviewer responses but shall reiterate again that we agree about
detection confidence; it would be optimal to do a validation in which we could quantify all true/false
positives/negatives, with an error matrix and associated statistics. However, due to the way the tool gets
a maximum debris extent using the image stacking method (if just one pixel in the image stack is debris, that pixel in the final mosaic will be debris), there is no dataset we can use to perform such a validation.
All existing datasets rely on an average or singular image to calculate debris coverage, which is
completely unsuitable for validating GERALDINE outputs against. We have confidence in outputs
though because the underlying image classification methods (cloud removal and band ratio algorithms)
work, as they have been used and peer-reviewed elsewhere. We have therefore undertaken a validation
in this way to provide some measure of RA detection accuracy and believe it is suitable for these
purposes.

*L228: But if topographic shading is classified as debris, it will influence new debris detection.*

Topographic shading is likely to be masked out of composites, as mentioned on L227, so it would not
influence new debris detection. If topographic shading was to be classified as new debris detection, as
mentioned above we do not mask it due to any artefact it produces (which is minimal), not displaying
supraglacial landslide characteristics and therefore not being flagged as a false positive. We found NDSI
to perform sufficiently well in shaded areas. In addition, it requires additional computational capacity
and subsequently increases analysis time for very little benefit with regards to supraglacial landslide
detection.

*L247: Your method has a high potential to detect all events [add: that are deposited onto initially bare*
*glacier ice]. E.g. a hypothetical second event at the same scarp on the glacier east of Maclaren Glacier*
*that deposited a slightly smaller volume of rock would be entirely missed by your method.*

And as mentioned in our response on lines 381-392, we shall add a sentence at L228 addressing multiple
failures; "GERALDINE can also not detect landslide debris deposition onto an existing debris cover.
Therefore if a supraglacial landslide consists of multiple failures, GERALDINE would only indicate
one event, with the deposit extent being the combined total of all failures." This is of course a problem
any manual identification of deposits will have from remote sensing. Running GERALDINE alongside
seismic detections has the best chance of resolving this, seismic noise will likely be high for a landslide
overrunning another rough, angular deposit.

*L250-256: I find this to be significant conditionality and required prior knowledge for an automated*
*tool. Your method doesn't automatically run for multiple years sequentially, so how would someone*
*new to the area know where to start? Reading your Figure 4 alone suggests the BRG RAs were*
*deposited between 2017 and 2018, this is misleading. The mapped Lituya RA in Fig. 5 also appears*
*patchy, should the logic of L254 be followed and this area be dismissed as erroneous?*

We believe it is unlikely anyone would find/use the tool without first seeing the manuscript. If they
have not seen the manuscript, we link and advise reading it on the welcome screen that greets a user
when they run GERALDINE. We shall change the figure caption of Fig. 4 to "d) 2018 erroneous tool
detection of Black Rapids glacier RA deposits, which were deposited as a cause of the 2002 Denali
earthquake (Jibson et al. 2006)." With regards to Fig. 5, the user has identified that a RA has occurred
here and can be confident of its down-glacier movement, as the leading edge is typical of a RA, being
both elongated and lobate. There is also no noise around the deposit leading edge which could
compromise these measurements. GERALDINE debris maps are no different to most other products
utilised at user discretion. This is just another example of how GERALDINE outputs can be used, and
we explain the errors associated with it.

*L257: While translated features are present in your output (also translated features from flow*
*instabilities, see Herreid and Truffer, 2016) and are scientifically useful, these are errors with respect*
*to your intended tool function. If you can automatically differentiate feature translation from feature*
*deposition then this can be a nice side component to your study, otherwise I think you need to treat this*
*as error. Herreid, Sam, and Martin Truffer. "Automated detection of unstable glacier flow and a*
*spectrum of speedup behavior in the Alaska Range." Journal of Geophysical Research: Earth Surface*
*121.1 (2016): 64-81.*

As mentioned previously, this is caused by the tool calculating a maximum debris extent, which we believe is the optimum method using this workflow, for detection of all supraglacially landslides (those deposited both in the accumulation and ablation zones). It is not the optimum method for accurately mapping a glaciers debris cover. It would be difficult to remove these translated features without inhibiting the performance and subsequent usability of the tool. See above comments and changes associated with line 277:

"We note other areas are flagged as 'new debris' in 2013 and 2014. These are typically where downwasting has occurred exposing more of the valley walls, or where there has been temporal evolution of the debris cover e.g. glacier flowline instabilities. These flow instabilities can cause double-counting of debris when larger time windows are specified (described further in Herreid and Truffer, 2015). Both processes subsequently cause false classification as 'new debris', however, neither display supraglacial characteristics, so it is highly unlikely a user would mistake them for one."

*L275: How does reduced ablation over one year around the ELA, where ablation rates are generally low, increase surface velocities?*

We should be clearer that we mean reduced ablation under the deposit, causing debris expansion. We shall amend to "We suggest that the higher RA deposit velocities between 2012 and 2013 are a result of the immediate response of the glacier to reduced ablation rates directly beneath the debris, causing an ice-pedestal to form, from which debris is redistributed through avalanching off the sides, expanding debris coverage (Reznichenko et al., 2011)."

*L281: RAs on bare glacier ice in ablation zones are easy to identify from one recent image and your method also requires manual inspection. Here I think you should highlight your tool's ability to potentially catch events in the accumulation zone that have only a very short residence time.*

We shall reword the conclusion introduction to "GERALDINE is the first free to use resource that can rapidly highlight new supraglacial landslide deposits onto clean ice for a user-specified time and location. It can aggregate hundreds of Landsat images, utilising every available cloud-free pixel, to create maps of new supraglacial debris additions. Using the output maps produced, GERALDINE gives an objective starting point from which a user can identify new debris inputs, eliminating the time-intensive process of manually downloading, processing and inspecting numerous satellite images. The method allows user identification of mass movements deposited in glacier accumulation zones, which have very short residence times due to rapid advection into the ice. This is a process that has not previously been quantified."

*L284: This is the first mention of 100% successful identification which should first appear in the results section, but I also think it is incorrect. By considering only true positive area, a map that is entirely "new debris additions" will also have a 100% successful identification rate but is clearly meaningless. You need to score your success against false positive and false negative area.*

100% detection accuracy post-1991 is mentioned on L185 but we shall make it clearer by rewording L185 to "False negatives all pre-date 1991 (Figure 3), giving 100% successful user identification post-1991" See the response (lines 973-993 of our responses) above as to why false positives and negatives were not calculated. We understand that a map of 100% new debris additions would also have a successful identification, but as can be seen from our examples in Figures 4 and 5 that this is clearly not the case.

---

## Editor Decision (ED1)

[revised manuscript text omitted]

We provide both good and bad examples of cloud mask performance in Figures S3 and S4, respectively. Figure S3 showcases the cloud masks ability to accurately mask cloud that is obscuring part of the Lamplugh RA, removing it from further analysis (Figure S3). However, occasionally it can suffer over debris cover in some areas (Figure S4), due to the optical and temperature similarities of the debris to cirrus clouds. This has similarly been found in

Antarctica with sunlit rock misclassification as cloud (Burton-Johnson et al., 2016). However, the image stack methodology used by GERALDINE helps to negate these cloud masking discrepancies.

[Figure]

**Figure S3: A) Original Landsat image (LC08_060019_20160729), B) Cloud masked Landsat image. Masking shows good ability to eliminate cloud pixels from scenes.**

[Figure]

**Figure S4: A) Original Landsat image (LC08_067016_20180704), B) Cloud masked Landsat image. Masking shows poor ability to eliminate cloud pixels from scenes, with misclassification of lighter debris as cloud.**

**3.0 Global distribution of validation RAs**

[Figure]

**Figure S5: Global distribution of RAs used for GERALDINE validation (48 in total).**

**4.0 GERALDINE User Guide**

The tool is freely available to use at (https://code.earthengine.google.com/ca49d1a7d06012f3e919fba5be6de4f3) but requires a Google account authorized to use Google Earth Engine (GEE), which is free of charge if used for research and educational purposes (sign up for Google account here: https://accounts.google.com/signup/v2/webcreateaccount?flowName=GlifWebSignIn&flowEntry=SignUp and register for GEE access here: https://earthengine.google.com/). Exporting of tool outputs requires a Google Drive account, which is complementary with the Gmail account required to sign up for GEE. The tool is open access and GUI (graphical user interface) driven. Tutorials on how to use Earth Engine are available at https://developers.google.com/earth-engine/ but here we will provide instructions on how to use our tool to detect supraglacial debris inputs.

**Step 1:**

[revised manuscript text omitted]